# Concentration-induced spontaneous polymerization of protic ionic liquids for efficient in situ adhesion

Jun Zhang[1], Xuan Zhou[1], Qinyu Hu[1], Kaijian Zhou[1], Yan Zhang[1], Shengyi Dong[2], Gai Zhao[3] & Shiguo Zhang[1]✉

The advancement of contemporary adhesives is often limited by the balancing act between cohesion and interfacial adhesion strength. This study explores an approach to overcome this trade-off by utilizing the spontaneous polymerization of a protic ionic liquid-based monomer obtained through the neutralization of 2-acrylamide-2-methyl propane sulfonic acid and hydroxylamine. The initiator-free polymerization process is carried out through a gradual increase in monomer concentration in aqueous solutions caused by solvent evaporation upon heating, which results in the in-situ formation of a tough and thin adhesive layer with a highly entangled polymeric network and an intimate interface contact between the adhesive and substrate. The abundance of internal and external non-covalent interactions also contributes to both cohesion and interfacial adhesion. Consequently, the produced protic poly(ionic liquid)s exhibit considerable adhesion strength on a variety of substrates. This method also allows for the creation of advanced adhesive composites with electrical conductivity or visualized sensing functionality by incorporating commercially available fillers into the ionic liquid adhesive. This study provides a strategy for creating high-performance ionic liquid-based adhesives and highlights the importance of in-situ polymerization for constructing adhesive composites.

Adhesives, as essential chemical products in modern life, have received extensive attention in recent years[1]. Great efforts have been devoted to establishing strong and durable adhesion to meet various requirements[2], while the simultaneous promotion and balance of cohesion and interfacial adhesion strength remain significant obstacles in developing high-performance adhesives[3–5]. In general, the cohesion of the adhesive is related to the mechanical and viscoelastic properties of the adhesive itself, while the interfacial adhesion depends largely on the energetic parameters of the adhesive on the substrate[6,7]. Cross-linking, which generates covalent or noncovalent interactions between the molecular chains of adhesives, is recognized as one of the most reliable approaches to increasing cohesion strength[8,9]. Especially, noncovalent interactions, such as hydrogen bonding (H-bonding), host–guest interactions, and ion–dipole interactions, are widely exploited[10–16]. In such cases, mechanical energy can be effectively dissipated through reversible cleavage and reforming of dynamic interactions[17,18]. However, in most cases, these interactions highly restrict the chain mobility during adhesion for non-in situ cured adhesives and inevitably result in a significant decrease in interfacial adhesion strength, thus deteriorating the resulting adhesion performance[9,19].

[1]College of Materials Science and Engineering, Hunan University, Changsha 410004, China. [2]College of Chemistry and Chemical Engineering, Hunan University, Changsha 410082, China. [3]State Key Laboratory of Mechanics and Control of Aerospace Structures, Nanjing University of Aeronautics and Astronautics, Nanjing 210016, China. ✉e-mail: zhangsg@hnu.edu.cn

Inspired by the natural adhesion phenomenon of organisms such as mussels and sandcastle worms, the in situ spontaneous polymerization offers a promising strategy to alleviate the trade-off between cohesion and interfacial adhesion. Generally, organisms secrete low-molecular-weight adhesive proteins with high fluidity that interact well with the substrate surface[20,21]. These precursor proteins can then be cured and solidified to improve cohesion (Fig. 1a)[22-24]. The 3,4-dihydroxy-L-phenylalanine (DOPA) moiety is found to be indispensable in these proteins for in situ adhesion[25,26]. The functionalization of traditional polymers through DOPA chemistry has emerged as a popular method for preparing high-performance adhesive materials[27-29]. However, DOPA-based adhesives have several inherent drawbacks, such as complicated synthesis and irreversible oxidizing cross-linking, which seriously limit their large-scale preparation and practical application[30]. Similar to the natural in situ adhesion phenomenon, commercial cyanoacrylate and epoxy adhesives exhibit noteworthy performance. Nonetheless, the use of these adhesives in large quantities poses significant characteristic odors and environmental issues and results in wastage of resources due to their volatile organic solvents or irreversible adhesion process. It is highly desired to design non-DOPA adhesives that combine the advantages of in situ spontaneous polymerization, low cost, ease of synthesis, and high adhesion strength.

In this work, we demonstrated that low-cost and high-performance protic poly(ionic liquid) (PPIL) adhesives could be simply prepared by neutralization of 2-acrylamide-2-methyl propane sulfonic acid (AMPS, 0.010–0.035 \$ g$^{-1}$) with hydroxylamine molecules (such as diethanolamine, DEA, 0.005–0.043 \$ mL$^{-1}$) with subsequent concentration and removal of the solvent (Fig. 1b). The hydroxylamine molecules ionized the sulfonate group, creating more extended molecular chains for PPILs and allowing for the formation of interconnected polymeric networks through dynamic intermolecular interactions such as H-bonding and electrostatic interactions. The in

situ spontaneous polymerization process, which was conducted in a solution with gradually increased monomer concentration, was beneficial for improving both cohesion and interfacial adhesion strength (Fig. 1c, d). The resulting PPILs had high molecular weight and highly entangled molecular chains, resulting in a maximum adhesion strength of up to 16.2 MPa. Furthermore, the PPIL adhesive was highly resistant to organic solvents and the adhesive process can be conducted directly in these solvents. The incorporation of thermo-chromic (TC) materials or multi-walled carbon nanotubes (MWCNTs) allowed for the creation of multifunctional adhesives with advanced applications such as visualized sensors and electrically conductive composites. Notably, these conductive composites facilitate the fabrication of electrodes for high-performance batteries.

## Results

### Synthesis and properties

PPILs were synthesized by stoichiometric mixing of the hydroxylamine molecules with AMPS in deionized water, followed by removal of the solvent upon heating. The neutralizing reaction of the monomer solution was confirmed by $^1$H NMR (Supplementary Figs. 1–6). Although storing the mixed solution at ambient temperature resulted in no discernible polymerization even after six months (Supplementary Fig. 7), spontaneous polymerization could be initiated by increasing the monomer concentration at elevated temperature in the absence of any additional initiator. This polymerization process of monomer solution was investigated using diethanolamine (DEA) as a hydroxylamine example. The thermogravimetric analysis (TGA) curve of the A2 monomer solution (denoted as A2ms) revealed a substantial weight loss upon heating from 30 to 110 °C, indicating the solvent evaporation (Fig. 2a). It should be noted that the neutralization reaction significantly inhibited the volatility and enhanced the thermal stability of the hydroxylamine molecule (Supplementary Fig. 8). The residual weight of A2ms consistently correlated with the solid content

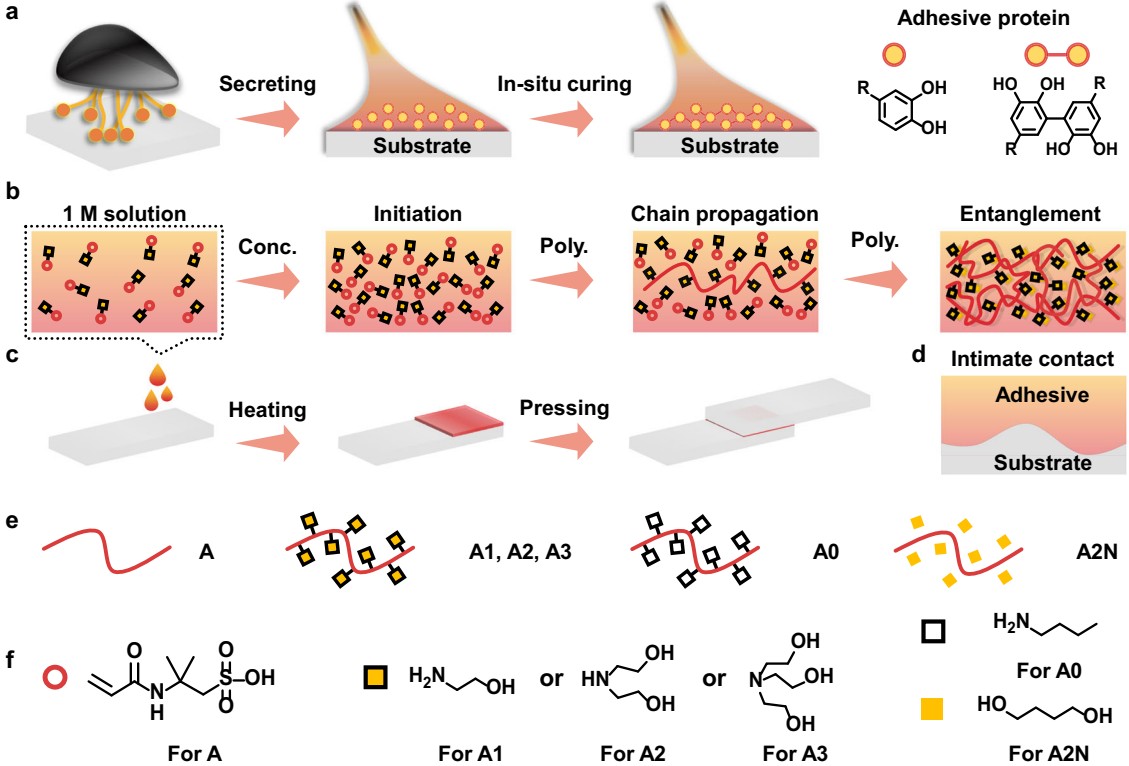

**Fig. 1 | Schematic of the in situ spontaneous polymerization of adhesive materials. a** Mussel-adhesion process, **b** the in situ polymerization of A2ms, **c** the in situ adhesion procedure using A2ms, **d** proposed interface region, **e** molecular chains of A2 and reference samples, and **f** structures of the AMPS, hydroxylamine molecules (ethanolamine, diethanolamine, and triethanolamine), and other organic molecular additives (1,4-butanediol, and n-butylamine) being investigated.

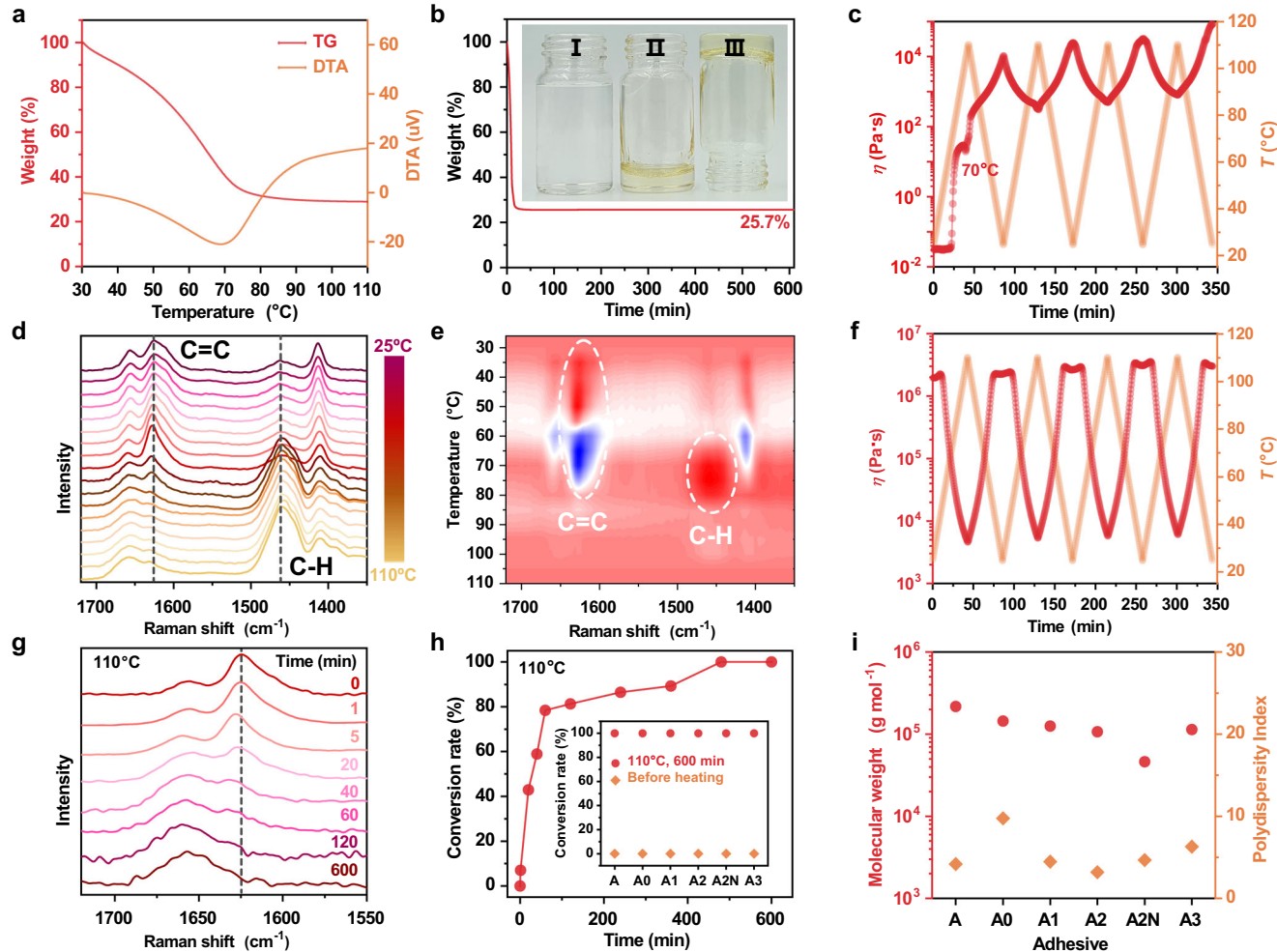

**Fig. 2 | Preparation and properties of the A2 and reference samples. a** TGA curves and **b** isothermal TGA of A2ms (concentration: 1 M). The inset in Fig. 2b shows the macroscopic state of A2ms before (I) and after isothermal curing treatment (II and III) (temperature: 110 °C, time: 600 min). The typical "inversion test" method (from II to III) showed that the resulting product was highly stuck on the bottom of the glass bottle without fluidity. **c** Reversible temperature-dependent rheological test of A2ms (angular frequency: 10 rad s$^{-1}$, strain: 1%). **d** Temperature-dependent Raman spectra of A2ms upon heating from 25 to 110 °C (interval: 5 °C). **e** Synchronous perturbation correlation moving-window spectra generated from (**d**), wherein red and blue colors are defined as positive and negative intensity, respectively. **f** Reversible temperature-dependent rheological test of the spontaneously polymerized A2 (angular frequency: 10 rad s$^{-1}$, strain: 1%). **g** Raman spectra and **h** conversion rate of A2ms with variable isothermal curing time. The inset in Fig. 2h shows the conversion rate of reference sample monomer solution before and after isothermal curing treatments. **i** Molecular weight and polydispersity index of the spontaneously polymerized A2 and reference samples obtained using other organic molecular additives.

ratio in a 1 M theoretical concentration aqueous solution (experimental weight ratio: 25.7%, theoretical weight ratio: 23.8%). Isothermal TGA curve further showed that the solvent evaporation was completed within the initial 20 min at 110 °C (Fig. 2b). The monomer solution underwent a process of gradual concentration and polymerization during heating, leading to a significant increase in viscosity, as depicted in Fig. 2c. Upon completion of the initial heating-cooling cycle (25 → 110 → 25 °C), there was a sharp rise in viscosity of five orders of magnitude. This substantial alteration in viscoelastic properties was distinguishable from that of the reference solution (1 M DEA) without AMPS monomer (as seen in Supplementary Fig. 9). In the following cyclic rheological test, a continuous increase in the viscosity of A2ms was observed, suggesting the occurrence of the polymerization reaction. To gain an insight into this spontaneous polymerization, temperature-dependent Raman spectra were collected (Supplementary Fig. 10). As indicated in Fig. 2d, e, the intensity of the peak at 1627.4 cm$^{-1}$ ($\nu_{C=C}$) progressively increased before reaching 70 °C, and then dropped rapidly. In contrast, the intensity of the peak at 1459.7 cm$^{-1}$ ($\delta_{CH2}$) exhibited the opposite transition. This result

indicated that hydrophobic vinyl groups were initially self-enriched in the concentration process and subsequently polymerized (Supplementary Fig. 11). After being subjected to isothermal curing at 110 °C for 600 min, polymeric materials with no fluidity were formed (Fig. 2b inset). The produced PPIL (A2) displayed a remarkable reversibility in viscosity upon changes in temperature, indicating the polymerization reaction was negligible, and the stability of the macromolecular network (Fig. 2f). The completion of polymerization after isothermal curing was confirmed by the disappearance of the characteristic peak of the vinyl group in the Raman spectra (Fig. 2g).

The polymerization efficiency of monomers was analyzed quantitatively using $^1$H NMR. As demonstrated in Fig. 2h and Supplementary Fig. 12, the conversion rate rapidly reached 78.4% after a 60-min curing period, exhibiting an auto-acceleration effect[31,32]. With further prolonging the curing time (e.g., 600 min), complete polymerization with a conversion of 100.0% was achieved. This spontaneous polymerization process was also observed in other solutions containing AMPS-based monomers and various organic molecular additives (Supplementary Figs. 1–6). Different from traditional solution

polymerization, the spontaneous process resulted in densely entangled PPILs with high molecular weight ($M_W$) and polydispersity index (PDI). Gel permeation chromatography confirmed that the average $M_W$ (>100 kDa) and PDI (>3) of the PPILs were both greater than that of the APS-A2 reference sample, which was synthesized through conventional polymerization using ammonium persulfate (APS) as the initiator (Fig. 2i and Supplementary Table 1). Moreover, the A2 PPIL exhibited higher moduli than the APS-A2 reference sample and a sample with partially disentangled molecular chains (DE-A2) (for details, see the Supplementary Information, Supplementary Fig. 13). During spontaneous polymerization, the monomer that is present in high concentrations tends to form a fabric-like structure within the pre-existing networks of the polymer (Fig. 1b)[33,34]. This structure, with its abundant entanglements, can significantly strengthen the PPILs, making them more cohesive[35–37]. Analysis using differential scanning calorimetry shows that the glass transition temperature ($T_g$) of the A2 PPIL decreased significantly from 103 °C for the nonionic reference sample A (homopolymer of AMPS) to 55 °C (Supplementary Fig. 14). The moderate phase transition temperature of A2 allows it to be mechanically processed at relatively low temperatures, such as 110 °C. For example, flexible and viscous fibers of A2 can be easily drawn when

separating two adhered plates at 110 °C (Supplementary Fig. 15). Like traditional poly(ionic liquid)s, A2 is nonflammable and cannot be ignited even when exposed to a flame (Supplementary Figs. 16, 17 and Supplementary Movie 1)[4]. Additionally, the spontaneous polymerization using low-cost precursors and non-organic solvents makes A2 very attractive for large-scale production. The process can easily be scaled up to produce one kilogram of A2 adhesive (Supplementary Fig. 18).

## Adhesion performance

By virtue of the in situ spontaneous polymerization, the substrate (e.g., ceramic) surface with A2ms was isothermally cured and was hot-pressed with another substrate at 110 °C (Fig. 1c). Upon cooling to room temperature, an adhesive interlayer of ~15 μm in thickness was formed with close and intimate contact between the two substrates (Fig. 3b inset and Supplementary Fig. 19). The macroscopic adhesion test demonstrated that even with a small adhesion area of 1 cm², a weight of 52 kg could be easily hung up without detachment (Fig. 3a and Supplementary Movie 2). Lap-shear tests were performed to quantitatively assess the adhesion strengths, yielding average adhesion strengths under ambient conditions on ceramic, epoxy (EP), and stainless steel (SS) of 15.6, 12.5, and 9.2 MPa, respectively (Fig. 3b),

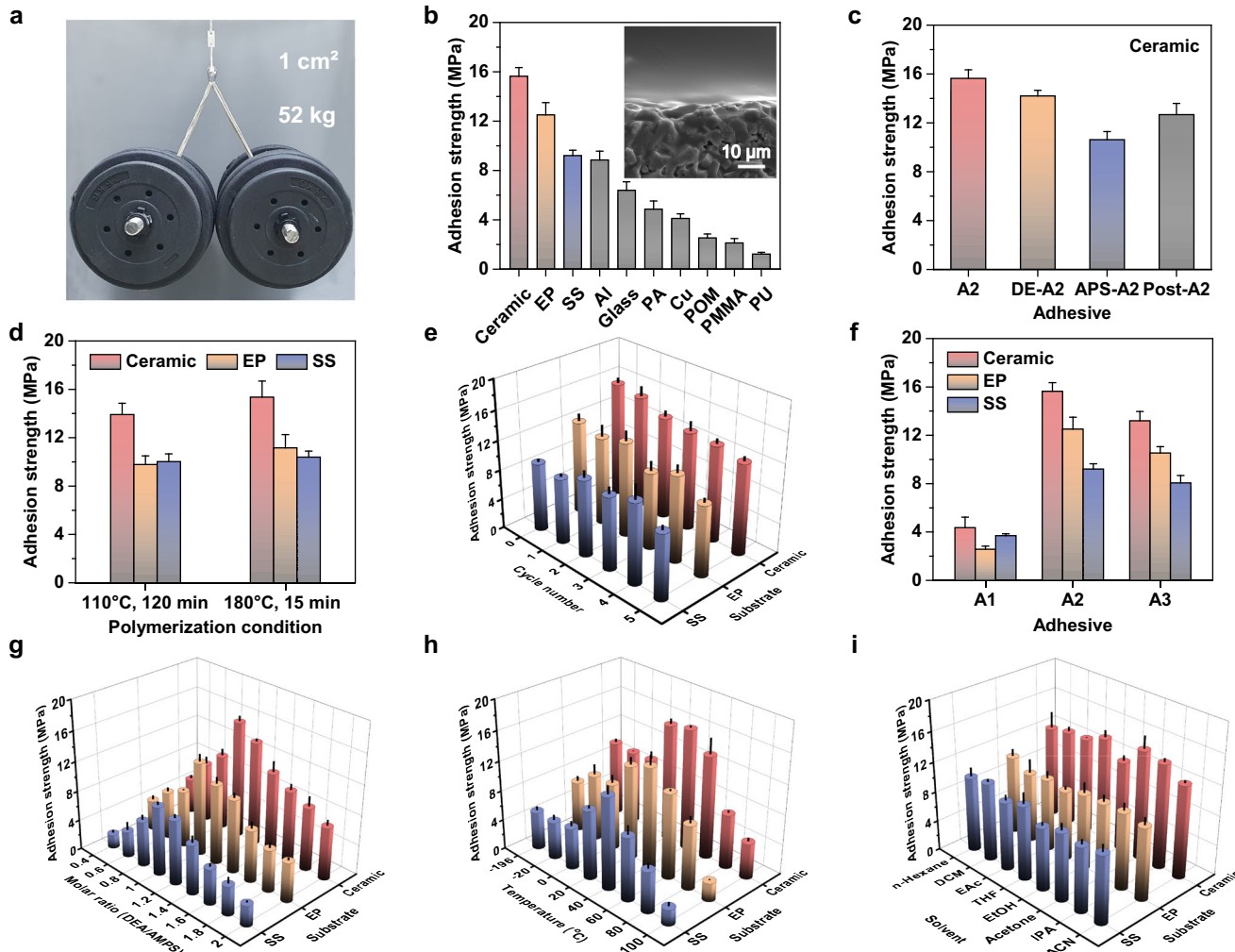

**Fig. 3 | Measurements of adhesion strength. a** Macroscopic adhesion test on ceramic substrates (weight: 52 kg, adhesion area: 1 cm²). **b** Adhesion strengths on various substrates at room temperature. The inset shows the SEM image of the interface between spontaneously polymerized A2 (top) and the ceramic substrate (bottom). **c** Adhesion strengths of A2 and other reference samples obtained by different processes. **d** Adhesion strengths of A2 obtained by different spontaneous polymerization conditions. **e** Successive cycling adhesion tests. Adhesion strengths of A2 and other reference samples obtained by **f** different hydroxylamine molecules and **g** different monomer molar ratios of DEA to AMPS. **h** Temperature-dependent adhesion strengths. **i** Adhesion strengths of A2 after soaking in various organic solvents for 1 month. All data were presented as mean ± SD ($n = 3$–5 independent samples).

which are much higher than most commercially available adhesives (Supplementary Fig. 20 and Supplementary Tables 2−5). Desirable adhesion strengths were also observed on glass (6.4 MPa), Al (8.9 MPa), Cu (4.1 MPa), polyamide (PA) (4.9 MPa), polyformaldehyde (POM) (2.5 MPa), polymethylmethacrylate (PMMA) (2.1 MPa), and polyurethane (PU) (1.2 MPa). The in situ spontaneous polymerization was responsible for the ultra-strong adhesion effect. The results in Fig. 3c demonstrated that the adhesion strength of A2 was significantly higher than that of the reference samples, including DE-A2, APS-A2, and post-A2 (post-A2 refers to using 1 M aqueous solution of pre-spontaneously polymerized A2 for adhesion test, for details, see the Methods).

The influence of polymerization conditions on adhesion strength was investigated. As shown in Fig. 3d, after undergoing an isothermal curing process at 110 °C for 120 min, the adhesion strengths on ceramic, EP, and SS surfaces were 13.9, 9.8, and 10.0 MPa, respectively. When the temperature was further increased to 180 °C, substantial conversion of monomers and high adhesion strengths could be achieved within 15 min, which is more efficient than UV-cured adhesives (Supplementary Fig. 21)[8,15]. Increasing the temperature, as shown by employing a higher-temperature laboratory heat gun (around 170 °C) for 2 min, markedly speeds up the polymerization reaction kinetics, resulting in a 100% conversion rate and substantial adhesion strength of 10 MPa (Supplementary Figs. 22−24). However, it is essential to note that excessively high processing temperatures pose safety hazards. To address this concern, adhesion strength measurements were conducted at safer curing temperatures between 50 and 100 °C. As shown in Supplementary Figs. 25, 26, even at low curing temperatures (50 to 80 °C), conversion rates of 100% of monomer were achieved. Following isothermal treatments, PPIL still exhibited a high adhesion strength ranging from 8.3 to 14.0 MPa (Supplementary Fig. 27). In addition, as shown in the repeated cycling adhesion tests, the A2 adhesive can be reused without obvious reduction in adhesion strength after five cycles (Fig. 3e). The adhesion effect depends on the chemical structure. Various hydroxylamine molecules could serve as raw materials for in situ polymerization, facilitating the preparation of high-performance adhesive materials (Fig. 3f and Supplementary Figs. 28, 29). Note that adhesive materials were only obtained from AMPS-based monomer solutions containing organic base molecules with hydroxyl groups. When hydroxylamine molecules were replaced by 1,4-butanediol or n-butylamine, the corresponding products (A2N and A0) were powder-like even at 110 °C and exhibited no adhesion toward any substrate (Supplementary Fig. 30). It is obvious that the hydroxylamine molecules resulted in the macroscopic transformation of brittle powder-like materials (A2N and A0) into strong and tough adhesives (A1−A3, and P1−P15). The relative mole ratio of DEA and AMPS affects the adhesion strength of the spontaneously polymerized mixture. The A2 adhesive, with an equimolar DEA and AMPS ratio, showed maximum adhesion strength as expected (Fig. 3g). These results suggested that hydroxylamine molecules provided efficient dynamic intermolecular interactions such as H-bonding and electrostatic interactions, causing great influences on the matrix and interface properties. Generally, most H-bonding interactions are heat-labile and could be spontaneously reconstructed at different temperatures[10], as evident from the temperature-dependent adhesion test. As shown in Fig. 3h, the maximum adhesion strengths of the A2 adhesive were observed at 40 °C on ceramic, EP, and SS surfaces, with values of 16.2 MPa, 13.6 MPa, and 12.3 MPa, respectively. This is because polymeric adhesives exhibit effective energy dissipation upon deformation near their $T_g$, resulting in improved adhesion performance[19]. In traditional single-lap joints, challenges arise from uneven stress distribution and interface stress concentration caused by variations in thermal expansion and mechanical properties with temperature changes. This tendency often results in the brittleness of the adhesive material, especially at low temperatures. Notably, even at the low temperature

of liquid nitrogen (−196 °C), the A2 adhesive exhibited significant adhesion strengths of 10.0, 5.0, and 6.8 MPa on ceramic, EP, and SS surfaces, respectively. The A2 adhesive was found to be highly resistant to organic solvents and stable in a range of common organic solvents, including n-hexane, dichloromethane (DCM), ethyl acetate (EAc), tetrahydrofuran (THF), ethanol (EtOH), acetone, isopropanol (IPA), and acetonitrile (ACN). Immersing the A2 adhesive in these solvents for a month did not result in any swelling, and no NMR signals of the polymeric adhesives were observed, as well as minimal impact on adhesion strength (Fig. 3i, Supplementary Figs. 31, 32, and Supplementary Table 6). For instance, the adhesion strength on the ceramic surface immersed in DCM remained at 12.8 MPa, retaining 82.1% of its original value (Fig. 3i). In contrast, commercially available adhesives, such as cyanoacrylate-based adhesive 3 M PR100 and two-component acrylic-based adhesive 3 M DP8005, swelled in DCM and showed no adhesion after 24 h. It should be noted that the adhesion strength of A2 adhesive may decline in humid conditions due to its water-solubility (Supplementary Fig. 33). However, methods like physical blending with hydrophobic PTFE, chemical structure design of PPIL, and coating design of PPIL-adhered joints can be applied to improve its stability in humid environments (Supplementary Figs. 34−41).

## Applications

Typically, the presence of a solvent layer between the adhesive and substrate disrupts noncovalent interfacial interactions, leading to weakened or non-existent adhesion under organic solvents[32,38]. However, PPILs, such as A2, possess multiple dynamic interactions and high resistance to organic solvents, making them suitable for use as adhesives under such conditions. Given that A2 has a glassy state at room temperature, a quantity of DEA was introduced into A2 (with a molar ratio of DEA to A2 at 0.2) as a plasticizer to improve the segmental mobility during adhesion (Fig. 4a). As shown in Fig. 4b, c, the adhesion effect was related to the curing time in organic solvents, and reliable adhesion strength was obtained within 2 h. Taking adhesion on ceramic submerged in EtOH as an example, it was found that as the curing time was extended from 0.5 to 2 h, the adhesion strength increased from 4.2 to 7.3 MPa. This demonstrates the limited migration and diffusion kinetics of the surface polymer chains. It is noteworthy that this level of adhesion surpasses most values reported for adhesion in organic solvents and is effective in both polar solvents such as EtOH and nonpolar solvents such as n-hexane[38].

The temperature-dependent adhesion effect of A2 allows the creation of a visually-sensitive adhesive by incorporating commercially available thermo-chromic (TC) materials (Fig. 4d). Microscopic images, including light microscopy and scanning electron microscopy (Supplementary Figs. 42, 43), illustrate the uniform distribution of TC materials in the adhesive matrix. The fuzzy interface between the filler and the matrix material validates the intimate microscopic interfacial contact between the filler and PPIL. As shown in Fig. 4e, f, a distinct color transition (from deep blue to light blue and rose red) was observed when the PPIL/TC adhesive composite was heated from −20 to 80 °C, accompanied by an increase in adhesion strength from 2.1 to 5.4 MPa. The variable temperature rheological test showed highly reversible viscoelastic behavior in the PPIL/TC composite. Additionally, the PPIL/TC composite displays stable and reversible color changes even after exposure to different temperatures for a week or undergoing ten cycles (Supplementary Figs. 44−46). This outcome realizes a cost-effective and non-destructive method of continuous visual monitoring of adhesion strength without the need for complex synthesis[39].

The in situ spontaneous polymerization makes it feasible to fabricate functional adhesive composites. A variety of inorganic fillers, such as MXene, MWCNTs, reduced graphene oxide (rGO), $Al_2O_3$, $Li_{1+x}Al_xTi_{2-x}(PO_4)_3$ (LATP), and hexagonal-boron nitride (h-BN), can be homogeneously incorporated into A2ms, followed by in situ spontaneous polymerization onto the substrate (Fig. 4g). The inclusion of a

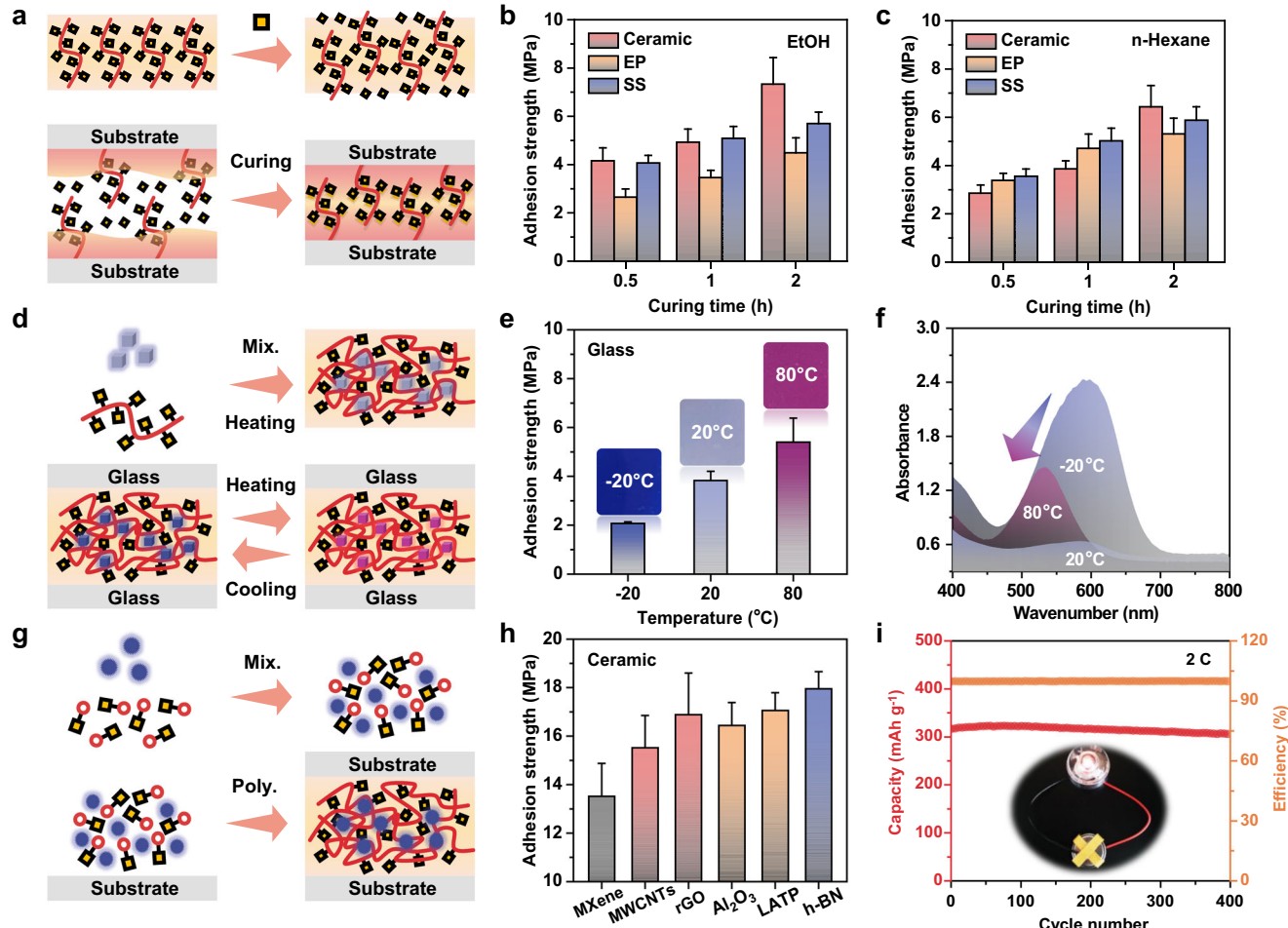

**Fig. 4 | Application of A2 adhesive materials. a** Schematic of the adhesion procedure under organic solvents. Top: using additional DEA as a plasticizer to enhance the mobility of surface polymer segments. Bottom: curing process under organic solvents. **b, c** Adhesion strengths of A2 under organic solvents of EtOH and n-hexane. **d** Schematic of preparation of A2 composite with thermo-chromic materials. **e** Adhesion strengths and **f** UV-vis spectra of A2 composites with thermo-chromic materials (content: 20 wt%) at different temperatures. The inset

photographs in **e** illustrate the color transition upon heating the adhesive composite from −20 to 80 °C. **g** Schematic and **h** adhesion strengths of the in situ prepared A2 composite with functional inorganic fillers. (i) Charge-discharge cycle performance of the graphite/Li cells at current densities of 2 C. The inset shows that LED lamps can be ignited using the graphite/Li cell that has been charge-discharge for 400 cycles. All data are presented as mean ± SD (n = 3–5 independent samples).

certain amount of fillers could endow the PPIL adhesives with functionalities without hindering the polymerization process or compromising adhesion strengths (Supplementary Fig. 47). For example, after in situ incorporating 10 wt% MWCNTs into the composite, the adhesion strengths on the ceramic substrate only slightly decreased from 15.6 MPa for the pristine A2 to 15.5 MPa (Fig. 4h). For PPIL/MWCNT composites, the well-preserved characteristic peaks (D and G) of MWCNTs suggest that the straightforward preparation process has minimal impact on MWCNTs, providing the composite adhesive with a high electrical conductivity of 1.53 S cm$^{-1}$ (Supplementary Figs. 48, 49). The in situ spontaneously polymerized PPIL/MWCNT adhesive not only retains pristine thermal stability but also demonstrates advantages in both adhesion strength and electrical conductivity compared to a reference composite produced using pre-polymerized A2 (12.7 MPa, 0.13 S cm$^{-1}$, Supplementary Figs. 49, 50). This high electrical conductivity is attributed to an optimized configuration of molecular chains and conductive pathways[40,41] (Supplementary Figs. 51, 52).

The highly conductive PPIL/MWCNT adhesive, serving as a binder/conductive additive composite component, demonstrates its utility in batteries. As an initial application, we employ it as a binder/conductive additive for a graphite anode in lithium-ion batteries. After uniformly mixing the aqueous monomer solution with MWCNTs and graphite,

in situ polymerization was conducted on the current collector surface. As demonstrated in Supplementary Figs. 53, 54, the resulting material exhibits robust adhesion to the current collector, maintaining exceptional cycle stability and rate performance in battery tests, even after 250 and 400 charge-discharge cycles at 1 C and 2 C rates, with discharge capacities of approximately 354.2 and 306.6 mAh g$^{-1}$, respectively (Fig. 4i and Supplementary Figs. 55, 56). Additionally, the PPIL/MWCNT adhesive showcases excellent performance as a cathode binder for lithium iron phosphate (LiFePO$_4$) (Supplementary Figs. 57, 58). After 35 cycles of charge-discharge cycling at the 0.2 C rate, the discharge capacity obtained is ~147.0 mAh g$^{-1}$ (Supplementary Fig. 59).

**Proposed mechanism for adhesion**

Given that achieving good adhesion properties requires simultaneous improvement in both cohesive and interfacial bonding strength, additional analysis focused on intermolecular chain interactions of PPIL and interactions between PPIL and substrates. Generally, the viscosity of a polymer solution reflects the intermolecular interactions in the presence of solvents[42,43]. As shown in Fig. 5a, the viscosity of the A2 polymer solution was found to be much higher than that of A0 and A2N, indicating that the presence of hydroxylamine molecules has a significant impact in constraining the movements of segments even in

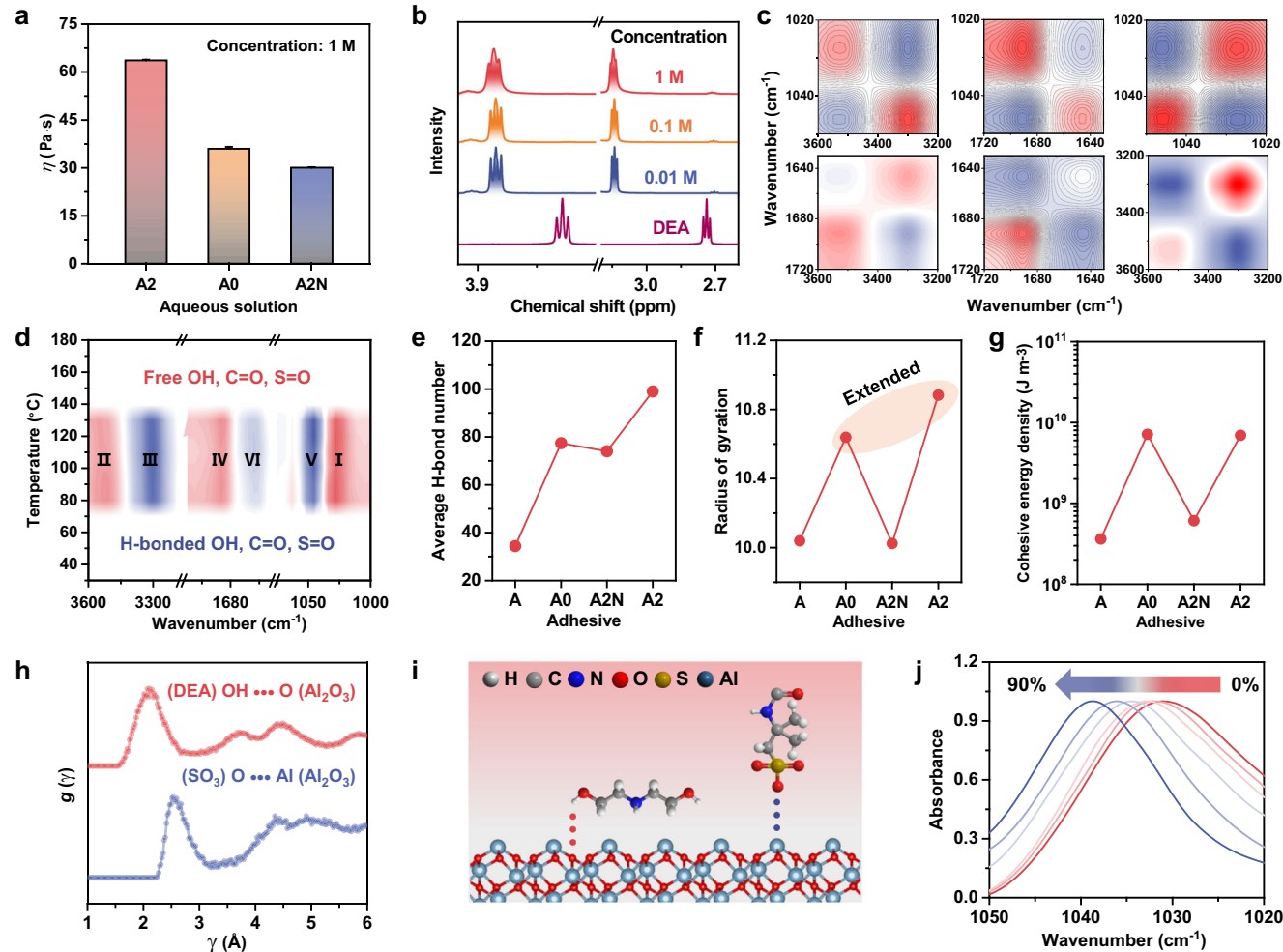

**Fig. 5 | Internal and external interactions of A2 adhesive materials. a** Solution viscosities of polymerized A2 and other reference samples (solvent: deionized water, concentration: 1 M). **b** Concentrations-dependent ¹H NMR spectra of solutions of spontaneously polymerized A2 (solvent: D₂O). **c** Two-dimensional COS synchronous spectra and **d** synchronous perturbation correlation moving-window spectra derived from temperature-dependent FTIR spectra. Red and blue colors in the contour maps of (**c**, **d**) are defined as positive and negative intensity, respectively. **e** Average H-bond number, **f** radius of gyration, and **g** cohesive energy

density of A2 and reference samples in the equilibrium state. **h** Radial distribution functions of (DEA) OH•••O (Al₂O₃) and (SO₃) OH•••Al (Al₂O₃) obtained from MD simulations. **i** Proposed interfacial interactions between A2 moiety and Al₂O₃ substrate. The interfacial H-bonding and electrostatic interactions are shown as red and blue dotted lines, respectively. **j** ATR-FTIR spectra of A2 composites with varying Al₂O₃ content (0 wt%, 10 wt%, 30 wt%, 50 wt%, 70 wt%, and 90 wt%). All data were presented as mean ± SD (n = 3–5 independent samples).

solution[11]. This effect becomes more pronounced at higher polymer concentrations, which can be analyzed through concentration-dependent ¹H NMR spectroscopy. Compared to pristine DEA, the CH₂ peaks in the PPIL solution shifted to low fields and broadened as the concentration increased, confirming the formation of intermolecular H-bonding and electrostatic interactions (Fig. 5b and Supplementary Fig. 60).

Temperature-dependent FTIR was conducted to investigate the dynamic interactions in the solid state. Upon heating from room temperature to 180 °C, H-bonded species were transformed into free states, as reflected by the decrease in intensity of the peaks at 1045 cm⁻¹ (H-bonded S=O), 1646 cm⁻¹ (H-bonded C=O), and 3301 cm⁻¹ (H-bonded O-H) and the increase in intensity of the peaks at 1028 cm⁻¹ (free S=O), 1689 cm⁻¹ (free C=O), and 3525 cm⁻¹ (free O-H) (Supplementary Fig. 61). To gain more information about the subtle variations, two-dimensional correlation spectroscopy (2DCOS) analysis was carried out. As shown in Fig. 5c, the auto peaks in the synchronous 2DCOS spectra at (1045, 1045 cm⁻¹) and (3301, 3301 cm⁻¹) are more pronounced than those in the (1028, 1028 cm⁻¹) and (3525, 3525 cm⁻¹) region, suggesting that H-bonded S=O and O-H groups are more

sensitive to temperature than the corresponding free groups when the temperature increases[44]. According to synchronous and asynchronous 2DCOS spectra, the order of response was as follows: 1028 → 3525 → 3301 → 1689 → 1045 → 1646 cm⁻¹ (with "→" means "earlier than", Supplementary Fig. 62 and Supplementary Table 7)[45,46]. This result suggested that as temperature increases, free S=O and O-H groups moved first and then exchanged with H-bonded O-H groups, followed by movements of free C=O, H-bonded S=O, and finally H-bonded C=O groups. Further analysis of the synchronous perturbation correlation moving-window (PCMW) spectra revealed that significant changes in the dynamic H-bonding interactions took place in the temperature range of 80 to 130 °C (Fig. 5d)[47]. This suggested that considerable H-bonded networks are formed at ambient temperatures, which was consistent with the temperature-sensitive viscoelastic properties of A2 (Fig. 2f).

Furthermore, molecular dynamics (MD) simulations revealed that the average H-bond number of A2 was higher than the reference samples of A, A0, and A2N, highlighting the important role of hydroxylamine molecules in constructing noncovalent interconnected networks (Fig. 5e and Supplementary Fig. 63). The radius of gyration,

which is a measure of the size of the polymer chain in a state of equilibrium[48], was found to be larger in A2 compared to the nonionic reference samples (A and A2N) due to the electrostatic repulsion effect of the ionized sulfonate group (Fig. 5f). The increased radius of gyration resulted in greater chain expansion and more interchain sites for enhanced intermolecular interactions. For nonionic A2N, even a considerable increase in the average H-bond number was observed, the limited radius of gyration indicated that H-bonding interactions were mainly formed in the 1,4-butanediol molecule domains. As expected, the cohesive energy density (CED) of A2 was found to be significantly higher ($6.95 \times 10^9$ J m$^{-3}$) than that of the nonionic reference samples A ($3.64 \times 10^8$ J m$^{-3}$) and A2N ($6.10 \times 10^8$ J m$^{-3}$) (Fig. 5g). This result can be attributed to the dominance of H-bonding and electrostatic interactions in the energy composition of CED (Supplementary Fig. 64). Although A0 had a slightly higher CED of $7.13 \times 10^9$ J m$^{-3}$, its limited average H-bond number impeded the formation of strong noncovalent interactions with external (e.g., substrate) surfaces, resulting in a brittle solid without any adhesion (Supplementary Fig. 30).

In addition to the bulk properties, the adhesion effect was highly influenced by the interfacial interactions between the PPILs and substrate. Taking ceramic substrate (using Al$_2$O$_3$ as a model) as an example, the radial distribution functions revealed the presence of both H-bonding ((DEA) OH•••O (Al$_2$O$_3$)) and electrostatic interactions ((SO$_3$) OH•••Al (Al$_2$O$_3$)) (Fig. 5h, i)[49,50]. Through physically mixing A2ms with Al$_2$O$_3$ and then undergoing spontaneous polymerization upon heating, A2 composites with varying Al$_2$O$_3$ contents were obtained. As shown in Fig. 5j, with increasing the Al$_2$O$_3$ content in the composite from 0 wt% to 90 wt%, the IR peak of the S=O peak noticeably shifted from 1031 cm$^{-1}$ for A2 to 1038 cm$^{-1}$ for composite with 90 wt% Al$_2$O$_3$. This blue shift indicated the presence of interactions (H-bonding and electrostatic interactions) between the S=O and the surface groups of Al$_2$O$_3$. These abundant noncovalent interactions and intimate interfacial contact played a significant role in increasing the overall strength of the interfacial adhesion. Eventually, a high interfacial adhesive energy (E$_{adhesion}$) of −1.45 Mcal mol$^{-1}$ was obtained in this system, comparable to previously reported values (Supplementary Fig. 65)[4,5].

## Discussion

In summary, we developed highly adhesive PPIL through a simple process involving the neutralization and spontaneous polymerization of hydroxylamine and vinyl group-containing acid in an aqueous solution and removing the solvent by heating. Temperature-dependent experiments revealed that the solvent evaporated most efficiently at 70 °C during the concentration process. At this transition temperature, vinyl group-containing IL monomers were self-enriched and subsequently polymerized, resulting in a significant increase in viscosity. Through isothermal curing, high molecular weight PPILs with highly entangled molecular chains were able to form in situ on the substrate, providing an intimate contact between the adhesive and substrate. Experimental analysis and theoretical calculations showed that an abundance of noncovalent interactions, such as H-bonding and electrostatic interactions, existed in the matrix and interface region, contributing to high cohesion and interfacial adhesion strength. The in situ spontaneously polymerized PPILs exhibited a maximum adhesion strength of 16.2 MPa, surpassing that of most DOPA-based adhesives and commercial adhesives. Even under harsh conditions such as ultra-low temperatures or prolonged exposure to organic solvents, no significant decrease in the adhesion strength was observed. Additionally, incorporating fillers into the PPILs allowed for the creation of adhesive composites with added functionality, such as visualized sensing and electrical conductivity. Given its elevated polymerization activity and resistance to environmental oxygen, PPIL-based composites with other properties like electromagnetic interference shielding, high mechanical strength, and thermal conductivity are anticipated to be easily produced in the future.

## Methods

### Chemicals, materials, and characterizations

2-Acrylamide-2-methyl propane sulfonic acid, ethanolamine, diethanolamine, triethanolamine, 1, 4-butanediol, n-butylamine, diethylaminoethanol, 2-(ethylamino)ethan-1-ol, 3-amino-1,2-propanediol, 2,2,2,2-ethylenedinitrilotetraethanol, 2-(cyclohexylamino)ethanol, N, N-dibenzylethanolamine, tris(hydroxymethyl)aminomethane, N-methyl-D-glucamine, N-phenyldiethanolamine, 8-aminooctan-1-ol, 2-(phenylamino)ethanol, 4-aminocyclohexan-1-ol, 1-deoxy-1-(n-octylamino)-D-glucitol, (4-(aminomethyl)phenyl)methanol, and (2 S,3 S,4 R)−2-aminooctadecane-1,3,4-triol were purchased from Adamas (China). Multiwall carbon nanotubes (MWCNTs) powder (4–6 nm outer diameter and 10–20 μm length) was purchased from Jiangsu XFNANO Materials Tech Co. Ltd. Graphite and lithium iron phosphate powder obtained from Aladdin (China) and Canrd Technology Co. Ltd (China) were used as active materials, respectively. MXene and rGO were prepared using previously reported methods (for details, see the Supplementary Methods)[51,52]. Ultra-pure purification system (Master-S15Q, Hitech Instruments Co. Ltd., Shanghai, China) was used to produce 18.2 MΩ cm$^{-1}$ water in all experiments. Al$_2$O$_3$, LATP, h-BN, thermochromic materials, and other organic reagents were commercially available. All chemicals used in this work were of analytical grade and were used as received without further purification.

$^1$H NMR spectra were collected on a Bruker 400 MHz with TMS as the internal standard. Fourier transform infrared (FTIR) and attenuated total reflectance infrared (ATR-FTIR) spectra were obtained on a Thermo Fisher Scientific Nicolet iS50 spectrometer equipped with a diamond single reflection attenuated total reflectance (ATR) accessory. Thermogravimetric (TG) analysis was obtained on a Hitachi STA7200 Thermal Analysis System at a heating rate of 10 °C min$^{-1}$ under an inert atmosphere of argon. Differential scanning calorimetry (DSC) was performed using a DSC822e Mettler-Toledo at a 10 °C min$^{-1}$ heating/cooling rate under an inert atmosphere of nitrogen (all values of $T_g$ were collected during the second heating cycle). Rheological measurements were processed on an Anton Paar MCR 92. The laminator was chosen with a diameter of 15 mm, a cone angle of 1°, and a gap of 0.051 mm. UV-Vis transmittance spectra were collected on a Shimadzu UV-2600 spectrophotometer. Raman spectra were collected using a DXR Raman Microscope with a 532 nm laser source. All morphologies of the PPIL/PTFE, PPIL/thermo-chromic materials, and PPIL/MWCNTs composite were characterized using a Hitachi-S4800 scanning electron microscope (SEM). All optical photographs (including reflection and transmission modes) were collected on an OptoNano 200 (Optosigma Corporation). A Bosch heat gun was used to accelerate the polymerization reaction kinetic process (the distance from the sample to the heat gun was controlled at 15 cm). The temperature of the sample under the laboratory heat gun was recorded by an IR thermal imager (Fluke TI450). Adhesion strengths were measured on an HY-0580 tension machine at a constant speed of 100 mm min$^{-1}$. Unless particularly noted, each test was carried out five times.

### Adhesion strength tests

A2 monomer solution (A2ms) with a concentration of 1 M (2.07 g AMPS, 1.05 g DEA, and 10 ml H$_2$O) was evenly coated on the surface of the substrate. After being thermal cured at 110 °C for 600 min, another pre-coated substrate was covered and hot-pressed. After pressing by a weight of 3 kg for 30 min at 110 °C, a thin adhesive layer (10 mm × 10 mm) about a thickness of 15 μm was formed, and two substrates adhered firmly. After creating adhesively bonded joints through in situ polymerization, we let them cure at room temperature (30–40% RH) for 3 h before assessing their adhesion performance.

The versatile in situ spontaneous polymerization process was proposed for preparing adhesive materials can be also applied to various structures.

 

## Traditional free radical polymerization of A2ms

Traditional free radical polymerization of A2ms (concentration: 1 M) was conducted at 80 °C in the presence of an initiator (1 mol %) of ammonium persulfate (APS). After polymerization, the resulting solution was dried under a vacuum to obtain the APS-A2. Afterward, APS-A2 was again dissolved in deionized water and stirred overnight at room temperature to obtain a polymer solution with a concentration of 1 M for adhesion.

## Disentanglement of spontaneously polymerized A2

A certain amount of spontaneously polymerized A2 was dissolved in deionized water and stirred for 1 week at room temperature to obtain a polymer solution with a concentration of 0.01 M. Subsequently, the solvent was evaporated at 110 °C under vacuum to obtain the DE-A2. Afterward, DE-A2 was again dissolved in deionized water and stirred overnight at room temperature to obtain a polymer solution with a concentration of 1 M for adhesion.

## Post-A2

Pre-spontaneously polymerized A2 was dissolved in deionized water and stirred overnight at room temperature to obtain a polymer solution with a concentration of 1 M for adhesion.

## The cycling adhesion strength test of A2 on various substrates

After the lap-shear test, substrates that cohesively separated were reheated at 110 °C and readhered. The adhesion area underwent additional pressing at 110 °C for 3 h. Subsequently, the prepared bonded joints were allowed to be set at room temperature (30–40% RH) for 3 h before conducting the adhesion performance test.

## Adhesion procedure under organic solvents

One substrate adhered by A2 was first immersed in organic solvents. Then, a certain amount of diethanolamine was cast on the first substrate surface, and followed by another substrate was covered and being pressed using a weight of 3 kg for different curing times.

## A2 composite with thermo-chromic materials

Typically, to prepare A2 composites with 20 wt% thermo-chromic materials, 3.12 g of spontaneously polymerized A2 was dissolved in 10 mL deionized water and mixed with 0.78 g thermo-chromic materials through 15 min of grinding. After 12 h of mechanical stirring, the solvent was evaporated at 110 °C on the substrate surface. Following thermal treatment at 110 °C for 30 min, another pre-coated substrate was hot-pressed. This process yielded a thin adhesive layer (10 mm × 10 mm) with a thickness of ~50 μm, firmly adhering to two substrates. Subsequently, after creating adhesively bonded joints through in situ polymerization, they were allowed to set at room temperature (30–40% RH) for 3 h before conducting an adhesion performance test.

## Preparation of A2 composite with inorganic or PTFE fillers

Typically, to prepare A2 composites with 10 wt% inorganic or PTFE fillers, 0.35 g of functional fillers, such as MWCNTs, was dispersed in 10 ml A2ms (concentration: 1 M) by grinding for 15 min. After 12 h of mixing and mechanical stirring, in situ, isothermal curing treatment (temperature: 110 °C, time: 600 min) occurred on the substrate surface. Subsequently, another substrate was hot-pressed, resulting in firm adhesion with a 25 μm adhesive layer. Electronic conductivity measurements of the A2/MWCNTs composite adhesive layer were conducted using a standard four-probe method on an RTS-8 four-probe instrument, with a reference composite fabricated using pre-spontaneously polymerized A2 for comparison. That is, 3.12 g of spontaneously polymerized A2 was dissolved in 10 ml deionized water and subsequently mixed with 0.35 g of MWCNTs fillers. For A2 composites with PTFE fillers, the hydrophobic nature necessitated over 40 min of grinding before mechanical stirring during preparation.

## Coating design of the PPIL-adhered joints

Petrolatum, derived from long-chain alkanes as a crude oil refining byproduct, is a white, uniform, and odorless ointment known for its high hydrophobicity. Coating modification was implemented using the adhesive obtained from in situ polymerization of the monomer aqueous solution on the substrate surface. Applying a thin layer of petrolatum at room temperature to the exposed areas of the PPIL in the bonding area gaps creates an effective waterproof membrane. This membrane efficiently prevents moisture diffusion in the air, mitigating potential adhesion failure at varying humidity levels.

## Electrode and cell fabrication

The slurry, comprising 80 wt% graphite or lithium iron phosphate, 10 wt% MWCNT, and 10 wt% binder (A2ms), was prepared in deionized water. Electrodes were formed by casting the slurry onto copper or aluminum foil using a doctor blade, followed by overnight drying under a vacuum at 80 °C. The dried electrodes, punched into 1.2 cm diameter disks, were used in coin cells for testing. 2025-coin cells were assembled for electrochemical testing, employing graphite electrodes as anodes, Li foil as counter/reference electrodes, and a polypropylene separator (25 μm, Celgard 2500). Electrolytes used were 1.0 M LiPF6 in 1:1 EC: DEC for graphite/Li cells and 1.0 M solvated ionic liquid (tetraglyme lithium bis(trifluoromethanesulfonyl)amide) for LFP/Li cells. The active material mass loading for graphite anodes and LFP cathodes is approximately 1.2 and 1.4 mg cm$^{-2}$, respectively. Half-cells were assembled in an argon-filled glovebox with moisture and $O_2$ content <0.1 ppm, and the assembled cells were allowed an 8-hour rest for stabilization before electrochemical characterization on the Neware battery testing system (BTS4000).

## Temperature-dependent Raman measurement of A2ms

About 20 μLA2ms was cast on a plate (diameter: 2 cm) with a temperature control accessory (Supplementary Fig. 10). For temperature-controlled measurement, the samples were heated from 25 to 110 °C with an interval of 5 °C, and each temperature holds for 2 min to reach equilibrium. The spectra range was from 3500 to 50 cm$^{-1}$ with 32 scans at a resolution of 4 cm$^{-1}$.

## Temperature-dependent FTIR measurement of A2

A certain amount of spontaneously polymerized A2 was dissolved in deionized water, subsequently, it was cast on a KBr tablet and heated at 100 °C under vacuum and sealed chamber with $CaF_2$ window for an hour before the test. An ultra-thin film of A2 on a KBr tablet was obtained for transmission IR measurement. The spectra range was from 4000 to 400 cm$^{-1}$ with 32 scans at a resolution of 4 cm$^{-1}$. For temperature-controlled measurement, the samples were heated from near room temperature of 30 to 180 °C with an interval of 10 °C, and each temperature was held for 10 min to reach equilibrium.

## Two-dimensional correlation spectroscopy (2DCOS)

The temperature-dependent Raman/FTIR spectra were used for analysis. 2DCOS and perturbation correlation moving-window (PCMW) spectroscopy were carried out using the software 2D Shige ver. 1.3 (Shigeaki Morita, Kwansei-Gakuin University, Japan, 2004–2005). In the contour maps, red colors are defined as positive intensities, and blue colors are defined as negatives. The responsive order of different groups can be judged by Noda's rule. The PCMW spectroscopy was applied to monitor the transition temperature. The temperature with the strongest intensity in the synchronous spectra can be utilized to determine the transition temperature range.

## Molecular dynamics simulation

In this work, molecular models of PPILs and $Al_2O_3$ ceramic were respectively built using Materials Studio software as shown in

Supplementary Fig. 65. Each PPILs model contains 15 molecular chains, of which each chain consists of 20 repeated monomers. Various molecular dynamics models were established initially, followed by the implementation of energy optimization processes for these models. Subsequently, dynamic trajectory and steady-state models for the molecular systems were obtained (Supplementary Data 1). The condensed-phase optimized potentials for atomistic simulation studies (COMPASS) force field was used. The COMPASS force field is commonly used to provide atomic interactions. To equilibrate the model, an equilibrating process was followed under constant temperature (25 °C) and constant volume (NVT ensemble) for 5 ns. During the simulation, the Nosé-Hoover thermostat was applied in the temperature control. For the most stable conformation with minimum energy models, the average H-bond number, radius of gyration, and cohesive energy density were analyzed. The restrictive geometry parameters of forming hydrogen bonding (H-bonding) interactions were a maximum hydrogen-acceptor distance of 2.5 Å and a minimum donor-hydrogen-acceptor angle of 90°. In order to investigate the dynamic process, the last 2 ns were analyzed. The average H-bond number was defined as the result of the total H-bond number divided by the number of molecular chains. The Ewald method and the atom-based method were employed for analyzing the Van der Waals interactions and electrostatic interactions between the adhesive molecules and the $Al_2O_3$ substrate surface. The radial distribution functions between adhesive molecules and $Al_2O_3$ were calculated based on the PPILs and the first $Al_2O_3$ layer. The cohesive energy density of adhesive molecules and interfacial adhesion energy between PPILs adhesive molecules and $Al_2O_3$ substrate surface were calculated by the following Eq. (1) and (2), respectively:

$$CED = E_{coh}/V \qquad (1)$$

$E_{coh} = -(E_{inter}) = (E_{intra}) - (E_{total})$, the brackets represent the time average.

$E_{coh}$: The cohesive energy.

$V$: The volume of a system.

$E_{inter}$: The total energy between all molecules.

$E_{intra}$: The intramolecular energy.

$E_{total}$: The total energy of a system.

$$E_{interfacial} = E_{total} - (E_{molecules} + E_{substrate}) \qquad (2)$$

$E_{interfacial}$: Interaction energy between PPILs adhesive molecules and $Al_2O_3$ substrate surface, minus means adsorption.

$E_{total}$: The total potential energy of a system.

$E_{molecules}$: The potential energy of PPILs adhesive molecules.

$E_{substrate}$: The potential energy of $Al_2O_3$ substrate.

## Data availability

The data that support the plots within this paper and other findings of this study are available from the corresponding author upon request.

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

## Acknowledgements

S.Z. acknowledges financial support from the National Natural Science Foundation of China (21872046, 52373206, and 52072118), the Jiebang Guashuai Project of Hunan Province (2021GK1230), the Research Found of YueLu Mountain Industrial Innovation Center (2023YCII0137), the Open Research Fund of Songshan Lake Materials Laboratory (2021SLABFN23), the Open Foundation of State Key Laboratory of Advanced Design and Manufacturing Technology for Vehicle (72275002), the Natural Science Foundation of Hunan Province (2020JJ4174). J.Z. acknowledges financial support from the Postgraduate Scientific Research Innovation Project of Hunan Province (CX20210404), the Natural Science Foundation of Hunan Province (2024JJ6141). J.Z. extends his gratitude to Shiyanjia Lab (www.shiyanjia.com) for providing invaluable assistance with the rheological analysis.

## Author contributions

J.Z. and S.Z. conceived and designed the experiments. J.Z. performed the experiments with help from X.Z., Q.H., K.Z., Y.Z., S.D., and G.Z. J.Z. and S.Z. wrote the paper. All authors analyzed the data, discussed the results, and commented on the manuscript.

## Competing interests

The authors declare no competing interests.
