## [Peer Review File · Nature Communications]

Concentration-induced spontaneous polymerization of protic ionic liquids for efficient in situ adhesionEditorial Note: Parts of this Peer Review File have been redacted as indicated to remove third-party material where no permission to publish could be obtained.

REVIEWER COMMENTS

Reviewer #1 (Remarks to the Author):

In this manuscript, the authors present a new approach to enhance the adhesion performance of structural adhesives by utilizing initiator-free polymerization with a chemically stable, cost-effective, and easily synthesizable protic ionic-liquid-based monomer. While the paper is well-organized, the central concept of "spontaneous polymerization for adhesives" is not entirely novel, as several other papers have previously reported similar ideas, including *Adv. Funct. Mater.* 2022, 32, 2109144; *Polymer* 2015, 64, 260-267; *Materials Science and Engineering: C* 2013, 33, 7, 3670-3676. Moreover, some perspectives mentioned by the authors in the introduction are not entirely accurate (outlined below). Additionally, the characterization of the adhesive preparation and the explanation of the underlying mechanism in the manuscript could be further refined. Given these observations, I recommend considering submission to other journals where the novelty and significance of their findings may be better appreciated.

Comments:

1. At the outset of this article, the authors emphasize the intriguing and worthy concept of the "cohesion-adhesion tradeoff." Through both experimental and calculated data, they demonstrate the significant contributions of H-bonding and electrostatic interactions to both cohesion and interfacial adhesion strength. However, it is regrettable that there are few convincing explanations provided to address this "tradeoff" issue in depth. The discussion regarding this "tradeoff" is confined to just the first paragraph of the introduction, and the corresponding reference (Yi et al. *ACS Appl. Polym. Mater.* 2021, 3, 2678) primarily focuses on pressure-sensitive adhesives. This raises a pertinent question: "Do adhesion and cohesion genuinely exhibit a strong negative effect on each other for in-situ adhesives?" I don't think so.
2. "Similar to the natural in-situ adhesion phenomenon, commercial cyanoacrylate and epoxy adhesives exhibit noteworthy performance. Nonetheless, the use of these adhesives in large quantities poses significant environmental issues and results in wastage of resources due to their volatile organic solvents and irreversible adhesion process." This conclusion is somewhat arbitrary, as there are a large number of commercially available water-based adhesives. No organic solvent, also strong adhesion.
3. Why 110 °C is chosen as the curing temperature throughout the test? How about other temperatures between 25 ~ 110 °C? If such high temperature is strictly required, this adhesive could only be applied in industrial scenarios. In fact, plenty of industrial in situ adhesives present a shear strength over 10 MPa.
4. In Figure 3e (SS substrate), the 2nd and 4th adhesion strength looks like higher than the original one. Such interesting result can be explained in the revised manuscript.
5. In the recipe of this adhesive material, AMPS and hydroxylamine are hydrophilic. Are all the samples in this work tested in dry state? Or how about the stability of these adhesives under a high-humidity condition?

Reviewer #2 (Remarks to the Author):

The work reports on a strategy to overcome the balancing between cohesion and interfacial adhesion strength in adhesives. The strategy is based on the spontaneous polymerization of a protic ionic liquid (IL)-based monomer obtained through the neutralization of 2-acrylamide-2-methyl propane sulfonic acid and hydroxylamine. A initiator-free polymerization process is presented leading to the in-situ formation of a tough and thin adhesive layer with a highly entangled polymeric network and a strong interface contact between the adhesive and the substrate. The method also allows the development of adhesive composites with electrical conductivity or sensing functionality by incorporating specific fillers. The work is in an area of strong and current interest, and in fact a novel strategy for the development of high-performance IL-based adhesive is presented.

The characteristics of the adhesive are convincingly demonstrated and the science behind the performance properly discussed, including computer simulation.

The work provides a mostly suitable description of the experimental work, allowing reproducibility in most cases. Most of the experimental data are properly discussed.

Nevertheless, several issues must be improved before publication of the work:

- adhesion assays were not performed following the standards (international norms) in the area. The authors should discuss why and how this affects potential applicability;

- the authors must provide adhesion performance parameters comparison with related compounds from the literature;

- the development of composites, both conductive and thermochromics is incomplete: important experimental information is missing, not allowing reproducibility and the characterization is incomplete in terms of morphological variations, filler dispersion and functional characteristics. This part must be extensively rewritten.

- There are no error bars and uncertainties in several parts on the work, even when the experimental variations are often within experimental errors.

Reviewer #3 (Remarks to the Author):

The contribution from Zhang et al on polymerized adhesives is interesting. I believe the study is well performed and thorough. The results are novel, though I am unsure there is a large advance over state of the art here. The use of ionic polymers in adhesives is not unknown, and though these are referred to as protic ionic liquids, the classification is dubious (they are fairly standard low molecular weight ionic polymers).

I think this is worthy of publication, but I would like a couple of items clarified:

1) What is the real advance here? I don't see what properties are much better than state of the art. The authors need to clarify and highlight this.

2) I don't think the lack of solvents is much of a sustainability advance, given the use of volatile organic amines in the production.

3) What evidence is there for conductive applications? I see it as possible, but as yet unverified.

The point-to-point response to the reviewers' comments

Reviewer #1 (Remarks to the Author):

In this manuscript, the authors present a new approach to enhance the adhesion performance of structural adhesives by utilizing initiator-free polymerization with a chemically stable, cost-effective, and easily synthesizable protic ionic-liquid-based monomer. While the paper is well-organized, the central concept of "spontaneous polymerization for adhesives" is not entirely novel, as several other papers have previously reported similar ideas, including *Adv. Funct. Mater.* 2022, 32, 2109144; *Polymer* 2015, 64, 260-267; *Materials Science and Engineering: C* 2013, 33, 7, 3670-3676. Moreover, some perspectives mentioned by the authors in the introduction are not entirely accurate (outlined below). Additionally, the characterization of the adhesive preparation and the explanation of the underlying mechanism in the manuscript could be further refined. Given these observations, I recommend considering submission to other journals where the novelty and significance of their findings may be better appreciated.

Response: We thank the reviewer for the evaluations of our work and constructive comments. The manuscript has been modified according to the reviewer's comments, and the changes are highlighted with a yellow background.

- First, we believe that the proposed concentration-induced spontaneous polymerization adhesion in our work is of great novelties and significance, mainly for the following reasons:

(1) Generally, the most crucial issue during the use of multi-phase materials is usually how to improve interfacial compatibility, and cutting-edge research in multiple fields beyond adhesion applications in recent years has shown that the in-situ curing process can effectively alleviate this problem (*Adv. Mater.* 2020, 32, 2001702; *Angew. Chem. Int. Ed.* 2023, 62, e202301241; *Angew. Chem. Int. Ed.* 2023, 62, e202309613; *Small Methods* 2021, 5, 2100505; *Adv. Mater.* 2023, 2308819; *Chem* 2023, 9, 2841). Regarding adhesion, despite the achievements in enhancing adhesive

cohesion, a new challenge must be addressed. Strong cohesion, essential for achieving adequate mechanical strain, frequently leads to poor interfacial adhesion, especially in non-in-situ adhesives. The in-situ spontaneous polymerization is effective and important for simultaneously increasing the cohesive energy and interfacial adhesion energy of adhesive materials. It is well known that mussels or sandcastle worms first produce an adhesive protein coacervate, which exhibits fluid-like properties, low interfacial energy, excellent spreading capacity, and interfacial bonding contact onto surfaces. After spreading and bonding, the adhesive could be cured and solidified quickly to improve its cohesion through covalent and non-covalent bonds. These natural systems have presented a captivating insight into realizing "in-situ" curing adhesives with simultaneously improved cohesive and interfacial strength of adhesive materials.

Some commercialized adhesives, such as those based on classic structures like cyanoacrylate, epoxy resin, and phenolic resin, can be considered in-situ adhesion to some extent. However, most commercial adhesives contain environmentally unfriendly organic solvents, and they are cross-linked structures that cannot be reused. Additionally, they have other drawbacks; for example, alkyl α -cyanoacrylate monomers require meticulous stabilization during manufacturing, storage, and transport to prevent premature polymerization due to their rapid setting characteristics at room temperature in the presence of moisture or weak bases (*Cyanoacrylate adhesives: A critical review. Reviews of Adhesion and Adhesives*, 2016, 4, 4). Acidic gases, including hydrogen fluoride, nitric oxide, sulfur dioxide, and sulfur trioxide, are employed in the manufacturing process to prevent premature polymerization by stabilizing the cyanoacrylate monomer (*Handbook of adhesive technology*, CRC Press, 2017, <https://doi.org/10.1201/978131512094>). Moreover, cyanoacrylate adhesives primarily use ethyl cyanoacrylate as their main component, a volatile monomer that can release monomer molecules into the air during drying, contributing to characteristic odors and presenting safety risks. Despite their typical small quantities, the concentration of odorant molecules in that volume can be relatively high, making the smell seem more potent. During curing, ethyl cyanoacrylate molecules react with moisture in the air, forming long polymer chains and releasing small amounts of odorous byproducts. Additionally, additives in cyanoacrylate glues, such as plasticizers, may contain volatile organic solvents, including esters like ethyl acetate or butyl acetate. Most cyanoacrylate adhesives also suffer from brittleness, and poor resistance to impact loads and organic solvents. Epoxy adhesives, despite the potential use of liquid precursors, also contain various additional volatile organic solvents like wetting agents, rheology modifiers, adhesion promoters, and plasticizers (*Epoxy Adhesive Formulations. McGraw Hill Professional, New York*, 2005, <https://doi.org/10.1036/0071455442>). Acknowledgedly, epoxy adhesives release volatile organic compounds, including acetone, benzene, and formaldehyde, during curing. Prolonged exposure to these substances may cause headaches, eye pain, respiratory tract irritation, and other health symptoms. **In contrast, we innovatively created poly(ionic liquid) adhesives using concentration-induced in-situ spontaneous polymerization of readily available precursors in aqueous solutions without any other agents (initiators, catalysts, cross-linkers). These**

adhesives offer unprecedented advantages, including environmental friendliness (no VOC emission), cost-effectiveness (approximately \$0.032 per gram for raw materials), high molecular weights exceeding 100,000, nearly 100% conversion rate with minimal impact from hydroxylamine structures, high adhesion strength reaching 16.2 MPa, outstanding performance at extremely low temperatures (10 MPa even at liquid nitrogen temperatures of $-196\text{ }^{\circ}\text{C}$), scalability to kilogram-level production, exceptional resistance to organic solvents, and the ability to create multi-functional composite materials, along with excellent battery performance surpassing most reported materials.

Our thorough examination of the literature on the rapid advancements in new adhesives revealed a scarcity of relevant concepts related to "in-situ spontaneous polymerization". Notably, there has been limited reporting of new structures in recent years. Upon careful scrutiny of the three references suggested by reviewers, we observed that, apart from the initial study (*Adv. Funct. Mater.* 2022, 32, 2109144) linked to spontaneous polymerization, the other two articles (*Polymer* 2015, 64, 260; *Materials Science and Engineering: C* 2013, 33, 7, 3670) are actually unrelated.

For the research paper of *Polymer* 2015, 64, 260-267: It was not related to research about spontaneous polymerization. In this work, Eriko Sato et al. reported that acetal-protected acrylic copolymers consisting of 1-isobutoxyethyl acrylate, 2-ethylhexyl acrylate, and 2-hydroxyethyl acrylate repeating units could be used in the application of dismantlable adhesives. The materials they used were synthesized by organotellurium-mediated radical polymerization. The significance of their research is that the material they prepared can be degraded after soaking in boiling water and under ultraviolet irradiation.

For the research paper of *Materials Science and Engineering: C* 2013, 33, 7, 3670-3676: Yong Wang et al. proposed an idea about spontaneous polymerization in self-etch dental adhesives, but their study lacks adhesion performance tests. The limited experimental data raises questions about the credibility of their findings. In essence, the impact of their research on the novelty and significance of our work is negligible.

Furthermore, our work differs significantly from the study reported in *Adv. Funct. Mater.* 2022, 32, 2109144. The specific distinctions are outlined below.

(a) In our study, following the isothermal treatment for solvent removal, monomers spontaneously transform into polymers with molecular weights exceeding 100,000. In contrast, the compared research reported an average molecular weight of only 1,400 for the polymerized materials after curing, as verified by gel permeation chromatography. This low molecular weight implies that the obtained polymer molecular chains consist of only a few dozen monomers. Consequently, the interaction between polymer molecular chains is too weak to produce elastic or glass-state polymers with high mechanical strength for a broad range of applications.

(b) The poly(ionic liquid) obtained in our research possesses tough, long-term, and recyclable adhesion on various substrates, including ceramic, epoxy, stainless steel, etc. The adhesion strength is up to 16.2 MPa, surpassing that of most catechol adhesives, commercially available adhesives, and existing ionic-based polymer adhesives. In contrast, whether it is a dry or underwater environment, the adhesion strength of the compared research is too low (just about 1 MPa), far from meeting basic requirements (even lower than several pressure-sensitive adhesives).

(c) The monomers we proposed in the in-situ spontaneous polymerization process are entirely different from those reported in the literature. The degree of conversion of the monomers into polymers is very high (almost 100%) and is little affected by types of hydroxylamine precursors. **Figure R2** shows that diverse hydroxylamine molecules can serve as raw materials for the synthesis of protic poly(ionic liquid) adhesive materials to achieve in-situ polymerization, enabling the preparation of high-performance adhesive materials. The polymerization reaction kinetic process can be significantly accelerated by elevating the temperature, demonstrated by the use of a higher-temperature laboratory heat gun (approximately 170 °C) for 2 min, yielding 10 MPa (**Figure R1**). This approach enables the rapid preparation of adhesive materials with a conversion rate of approximately 100%. In contrast, the spontaneous polymerization in the compared research shows low activity, with the conversion rate consistently below 80%, even with a prolonged reaction time of up to 35 hours. Besides, a slight change in their raw material structure would deteriorate the polymerization activity. The scalability that applies to other particular fields is seriously limited. The carcinogenic nature of residual 4-amino styrene monomers also poses a safety threat to operators.

(d) The neutralization reaction and spontaneous polymerization of our research are based on low-cost precursors (about 0.032 \$ g⁻¹ for raw materials), making it easily scalable for the preparation of kilograms of macromolecules. However, the raw materials used in the compared research are expensive (at least 10 \$ g⁻¹ for raw materials), have a strong irritating odor, and easily cause skin irritation and cancer (especially for the 4-vinyl aniline component).

(2) In addition to the abovementioned issues, our research also exhibits other novelties and significances:

(a) This study represents the first documentation of achieving nearly 100% yield in the straightforward preparation of poly(ionic liquid) through a one-pot neutralization reaction and concentration-induced in-situ spontaneous polymerization. Considering the widespread attention and promising applications of poly(ionic liquid) materials in various fields, coupled with the typically tedious synthesis and relatively low yields associated with most poly(ionic liquid)s in previous work, our work introduces a simple, efficient, and scalable synthesis strategy. This approach is poised to significantly advance the research and application of ionic liquid functional materials.

(b) Traditionally, the adhesive in a single lap joint faces challenges such as uneven stress distribution and interface stress concentration due to differing thermal expansion and mechanical properties with temperature variations. This often leads to the brittleness of the adhesive material, particularly at low temperatures. Our study reveals that in-situ polymerization effectively mitigates this issue. Notably, even at the ultra-low temperature of liquid nitrogen ($-196\text{ }^{\circ}\text{C}$), adhesion strengths of 10.0, 5.0, and 6.8 MPa were observed on ceramic, epoxy (EP), and stainless steel (SS) surfaces, respectively. As listed in **Table R1**, the low-temperature adhesion strength of the PPIL adhesive surpasses that of most currently reported adhesive materials. Currently, our group is actively researching ultra-strong low-temperature adhesion through an in-situ curing process. The results of this ongoing research will be published elsewhere.

(c) The in-situ spontaneous polymerization process of our research can be used to prepare functional composite materials with excellent comprehensive properties for diverse applications across various fields. As discussed in the response to Reviewer #3's third question, we illustrate one application: serving as a binder component in electrodes for lithium ion batteries. In this application, we performed in-situ polymerization on the copper foil surface after uniformly blending the aqueous monomer solution with carbon nanotubes and graphite. The resulting material exhibited robust adhesion to the current collector even after prolonged immersion in the electrolyte, demonstrating outstanding cycle stability and rate performance in battery tests. These results surpassed the performance of most reported materials in similar studies (*Adv. Funct. Mater.* 2023, 33, 2302951; *ACS Energy Lett.* 2023, 8, 1336; *J. Power Sources* 2019, 429, 67; *ACS Sustainable Chem. Eng.* 2022, 10, 12023; *Chem. Eng. J* 2023, 476, 146299). In addition to serving as a binder for a graphite anode, the obtained poly(ionic liquid) exhibited excellent performance as a cathode binder for lithium iron phosphate (LiFePO_4 , LFP). Furthermore, composite adhesives with visual sensing functionality were created through the incorporation of fillers. Notably, the adhesive composite exhibited a distinct color transition, shifting from deep blue to rose red, as the temperature increased from -20 to $80\text{ }^{\circ}\text{C}$. This color change correlated with an increase in adhesion strength from 2.1 to 5.4 MPa. The integration of this visual sensing capability allows for non-destructive and continuous monitoring of adhesion strength, providing a cost-effective and straightforward alternative to labor-intensive synthesis methods.

(d) Due to their remarkable resistance to organic solvents, joints adhered with poly(ionic liquid)s remained insoluble and stable in a range of common organic solvents, including n-hexane, dichloromethane, ethyl acetate, tetrahydrofuran, ethanol, acetone, isopropanol, and acetonitrile. To address the reviewer's concerns, we meticulously compared the organic solvent resistance of PPIL with other reported functional adhesives and the relevant data are presented in **Table R2**.

- Secondly, regarding the question of "some perspectives mentioned by the authors in the introduction are not entirely accurate (outlined below)", we would respond to them point-by-point in the detailed response below.

- Finally, the detailed responses regarding the question "Additionally, the characterization of the adhesive preparation and the explanation of the underlying mechanism in the manuscript could be further refined" are listed as follows.

(1) In this research, we employed TGA, isothermal TGA, reversible temperature-dependent rheological testing, temperature-dependent Raman spectra, synchronous perturbation correlation moving window spectra, ¹H NMR, GPC, and DSC techniques to comprehensively investigate the adhesive preparation. By neutralizing hydroxylamine and vinyl group-containing acid in an aqueous solution and subsequently removing the solvent through heating, we achieved the in-situ spontaneous polymerization of highly adhesive PPILs. The solvent evaporated most intensively at 70 °C during the concentration process, leading to the self-enrichment and subsequent polymerization of vinyl group-containing ILs monomers. This resulted in a substantial (2 ~ 3 orders) increase in viscosity around this transition temperature. Following isothermal curing treatment, high molecular weight and highly entangled PPILs were formed in situ on the substrate, establishing intimate interface contact. Then, we conducted a lap shear test to evaluate the adhesion performance of the material. The in-situ spontaneously polymerized PPILs demonstrated a remarkable adhesion strength of 16.2 MPa, surpassing that of most DOPA-based and commercial adhesives. The adhesion strength remained consistently high even under ultra-low temperatures and prolonged exposure to various organic solvents.

(2) We systematically examined the adhesive mechanism through a comprehensive approach involving concentration-dependent ¹H NMR analysis, temperature-dependent FT-IR spectra, and theoretical calculations. Our investigation revealed the presence of in-situ formed highly interconnected polymeric networks characterized by abundant hydrogen bonding and electrostatic interactions in both the matrix and interface region. These interactions contribute to enhanced and balanced cohesion and interfacial interactions. We believe that these results have potential significance for explaining the adhesion phenomenon of ionic-based polymer adhesives.

(3) Owing to the robust activity of spontaneous polymerization, we achieved the in-situ fabrication of a multifunctional composite by incorporating organic or inorganic fillers (e.g., MWCNTs). The adhesion strength and electrical conductivity of composites produced through in-situ polymerization surpassed those obtained through conventional methods. Furthermore, the in-situ fabrication of electrically conductive composites was successfully applied to construct a composite binder for lithium-ion battery electrodes, demonstrating exceptional cycle stability and rate performance.

In order to address the reviewer's concern, the corresponding results are provided in the revised Supplementary Information (Supplementary Figs. 22 - 24, Supplementary Figs. 28 and 29, and Supplementary Figs. 53 -59). The corresponding description in the revised manuscript was modified as follows: *"Nonetheless, the use of these adhesives in large quantities poses significant characteristic odors and environmental issues and results in wastage of resources due to their*

volatile organic solvents or irreversible adhesion process”, “Increasing the temperature, as shown by employing a higher-temperature laboratory heat gun (around 170 °C) for 2 minutes, markedly speeds up the polymerization reaction kinetics, resulting in a 100% conversion rate and substantial adhesion strength of 10 MPa (Supplementary Figs. 22 - 24)”, “In traditional single lap joints, challenges arise from uneven stress distribution and interface stress concentration caused by variations in thermal expansion and mechanical properties with temperature changes. This tendency often results in the brittleness of the adhesive material, especially at low temperatures. Notably, even at the extremely low temperature of liquid nitrogen (−196 °C), the A2 adhesive exhibited significant adhesion strengths of 10.0, 5.0, and 6.8 MPa on ceramic, EP, and SS surfaces, respectively”, “The adhesion effect depends on the chemical structure. Various hydroxylamine molecules could serve as raw materials for in-situ polymerization, facilitating the preparation of high-performance adhesive materials (Fig. 3f, and Supplementary Figs. 28 and 29)”, and “The highly conductive PPIL/MWCNT adhesive, serving as a binder/conductive additive composite component, demonstrates its utility in batteries. As an initial application, we employ it as a binder/conductive additive for a graphite anode in lithium-ion batteries. After uniformly mixing the aqueous monomer solution with MWCNTs and graphite, in-situ polymerization was conducted on the current collector surface. As demonstrated in Supplementary Figs. 53 and 54, the resulting material exhibits robust adhesion to the current collector, maintaining exceptional cycle stability and rate performance in battery tests, even after 250 and 400 charge-discharge cycles at 1 C and 2 C rates, with discharge capacities of approximately 354.2 and 306.6 mAh g^{−1}, respectively (Fig. 4i, and Supplementary Figs. 55 and 56). Additionally, the PPIL/MWCNT adhesive showcases excellent performance as a cathode binder for lithium iron phosphate (LiFePO₄) (Supplementary Figs. 57 and 58). After 35 cycles of charge-discharge cycling at the 0.2 C rate, the discharge capacity obtained is approximately 147.0 mAh g^{−1} (Supplementary Fig. 59)”.

Figure R1. In-situ polymerization of A2 using a higher-temperature laboratory heat gun. (a) Adhesion strengths of A2 with different initial monomer concentrations obtained by polymerization conditions using a higher-temperature laboratory heat gun (roughly 170 °C) for 2 min. (b) The temperature detected by an IR thermal imager for ceramic substrate with A2 under a laboratory heat

gun for 2 min.

Table R1. Comparison of adhesion strength of PPILs and other reported functional adhesives at low temperatures.

Adhesive	Temperature (°C)	Adhesion strength (MPa)	Reference
PC10-W1	-196	1.2	J. Am. Chem. Soc. 2020, 142, 21522
TA-epoxy	-196	0.6	Biomacromolecules 2022, 23, 3493
CA/PEG2000	-196	0.9	ACS Appl. Polym. Mater. 2022, 4, 4319
Poly(PA-H)	-60	0.4	ACS Appl. Mater. Interfaces 2022, 14, 27476
C12TAB/ChCl-urea	-80	1.0	Mater. Horiz. 2022, 9, 1700
BSA0.35	-196	9.5	Adv. Sci. 2022, 220318
P1	-196	4.8	ACS Nano 2022, 16, 5303
NIPA5.0	-196	15.9	Chem. Mater. 2023, 35, 7730
6-HTPB	-80	2.1	Green Chem. 2023, 25, 6845
PEA	-196	5.6	Eur. Polym. J. 2023, 198, 112387
Poly(A-C)	-196	9.5	Chem. Eng. J. 2023, 451, 138674
Poly(TA-DB)	-65	7.9	Chem. Eng. Sci. 2023, 281, 119164
cPA2	-196	17.4	J. Mater. Chem. A 2023, 11, 6286
DPETI	-196	2.2	Adv. Mater. 2023, 2310779
A2	-196	10.0	This work

Table R2. Comparison of organic solvent resistance of PPILs and other reported functional adhesives.

Adhesive	Types of organic solvents	Soaking time	Adhesion strength after soaking(MPa)	Reference
VPTA	2	22 d	N/A	J. Mater. Chem. A 2017, 5, 21169
DPU-HMA	1	18 h	2.7	Mater. Chem. Front. 2019, 3, 1833
CT-2	5	21 d	2.7	CCS Chem. 2020, 2, 1690
SP-DN	7	24 h	0.8	Mater. Horiz. 2021, 8, 2520
P4-AS-PAA	2	6 m	0.9	Adv. Funct. Mater. 2021, 2109144
HPU-HMA	3	30 d	N/A	Ind. Eng. Chem. Res. 2021, 60, 6925
P1	9	100 d	4.8	ACS Nano 2022, 16, 5303
C12TAB/ChCl-urea	4	N/A	0.6	Mater. Horiz. 2022, 9, 1700
BSA0.35	3	24 h	14.6	Adv. Sci. 2022, 9, 2203182
PVA-PTA	2	30 min	N/A	Adv. Funct. Mater. 2022, 2111892
PEG-TA	2	7 d	4.0	Macromol. Rapid Commun. 2022, 43, 2100830
NIPA5.0	9	168 h	18.8	Chem. Mater. 2023, 35, 7730
6-HTPB	4	4 h	1.1	Green Chem. 2023, 25, 6845
A2	8	1 m	12.8	This work

Figure R2. Adhesion strengths of PPILs obtained by different hydroxylamine molecules.

Detailed responses are listed as follows.

1. At the outset of this article, the authors emphasize the intriguing and worthy concept of the "cohesion-adhesion tradeoff." Through experimental and calculated data, they demonstrate the significant contributions of H-bonding and electrostatic interactions to cohesion and interfacial adhesion strength. However, it is regrettable that there are few convincing explanations provided to address this "tradeoff" issue in depth. The discussion regarding this "tradeoff" is confined to just the first paragraph of the introduction, and the corresponding reference (Yi et al. ACS Appl. Polym. Mater. 2021, 3, 2678) primarily focuses on pressure-sensitive adhesives. This raises a pertinent question: "Do adhesion and cohesion genuinely

exhibit a strong negative effect on each other for in-situ adhesives?" I don't think so.

Answer: Thank you for this comment.

- According to established adhesion theory, both cohesive strength and interfacial adhesion strength serve as pivotal indicators of adhesion performance (*Adhesion and adhesives technology: an introduction*. Carl Hanser Verlag GmbH Co KG, 2021. ISBN: 978-1-56990-855-6; *Handbook of adhesion technology*. Springer Science & Business Media, 2011. <https://doi.org/10.1007/978-3-642-01169-6>; *Structural Adhesive Joints: Design, Analysis, and Testing*. John Wiley & Sons, 2020. ISBN: 978-1-119-73643-1). Cohesive strength pertains to the interaction between molecules within a material, influencing its overall strength and toughness. Higher cohesive strength indicates increased resistance to external tensile or shear damage, and this property is influenced by factors such as molecular structure, molecular weight, cross-linking density, and intermolecular forces. On the other hand, interfacial adhesion strength refers to the bonding force between adhesive material and other substances, playing a crucial role in determining the material's adhesion and durability. Greater interfacial adhesion strength results in a firmer bond to other substances, reducing the likelihood of delamination or peeling. This property is influenced by surface properties, wettability, adhesive selection, and material surface treatment.
- Cohesive and interfacial adhesion strength have distinct effects on adhesion performance, yet both are indispensable. For any adhesive (including reactive epoxy, polyurethane, acrylic, and silicone-based adhesives), its macroscopic adhesion performance tends to be constrained by one of the above factors. Adhesive failure (between the substrate and adhesive layer) or cohesive failure (within the adhesive layer itself) is observed in certain adhesive materials. That is, as long as there is a numerical value for adhesion strength, it is bound to be limited by one of the two sizes (**Figure R3 top**). Otherwise, this means that the adhesion strength cannot be measured (N/A for substrate failure). Consequently, achieving good adhesion properties necessitates simultaneous improvement in both cohesive and interfacial bonding strength. Thus, the central challenge in this field, applicable to various adhesives such as pressure-sensitive, hot-melt, solvent-based, and in-situ cured adhesives, is finding a balance and enhancing the interaction force between cohesive and interfacial components simultaneously (*Soft Matter*, 2019, 15, 3807; *Macromol. Chem. Phys.* 2023, 224, 2200332; *Adv. Mater.* 2023, 35, 2300802; *Adv. Mater.* 2023, 2310779; *Adv. Mater.* 2023, 2310576).
- Balancing interfacial adhesion strength and cohesion, which often exhibit an inverse relationship (**Figure R3 bottom**), is referred to as the "trade-off" effect. For non-in-situ cured adhesives, increasing cohesion can lead to rigidity with reduced interfacial interaction, while boosting interfacial adhesion strength may result in soft adhesives with weakened cohesion. Therefore, achieving an adhesive with optimal strength and flexibility requires meticulous optimization to strike the right balance between these properties.

In-situ spontaneous polymerization is an effective and crucial strategy for concurrently enhancing

the cohesive energy and interfacial adhesion energy of adhesive materials. However, it should be noted that even with the in-situ polymerization approach, it is not possible to simultaneously achieve an infinite increase in both cohesive energy and interfacial adhesion energy. There will inevitably be a situation where one of the two, either cohesive energy or interfacial adhesion energy, is lower than the other, resulting in the persistence of either cohesive or interfacial failure.

In fact, in the original manuscript, the description of “trade-off” between cohesive strength and interfacial adhesion strength was mainly confined to the widely researched conventional non-in-situ cured adhesives. In order to address the reviewer’s concern, the corresponding description in the revised manuscript was modified as follows: *“However, in most cases, these interactions highly restrict the chain mobility during adhesion for non-in-situ cured adhesives and inevitably result in a significant decrease in interfacial adhesion strength, thus deteriorating the resulting adhesion performance”, “Inspired by the natural adhesion phenomenon of organisms such as mussels and sandcastle worms, the in-situ spontaneous polymerization offers a promising strategy to alleviate the trade-off between cohesion and interfacial adhesion”, and “Given that achieving good adhesion properties requires simultaneous improvement in both cohesive and interfacial bonding strength, additional analysis focused on intermolecular chain interactions of PPIL and interactions between PPIL and substrates”.*

Figure R3. Schematic of different failure types during lap shear test and cohesion-adhesion trade-off for adhesives.

2. “Similar to the natural in-situ adhesion phenomenon, commercial cyanoacrylate and epoxy adhesives exhibit noteworthy performance. Nonetheless, the use of these adhesives in large quantities poses significant environmental issues and results in wastage of resources due to their volatile organic solvents and irreversible adhesion process.” This conclusion is somewhat arbitrary, as there are a large number of commercially available water-based adhesives. No organic solvent, also strong adhesion.

Answer: Thank you for this comment.

- The preceding statement underscores the similarity to the application of cyanoacrylates and epoxy adhesives, where the interfacial wetting ability of the precursor monomer solution and a sufficient degree of freedom (i.e., the capability to establish substantial physical-mechanical

interlocking and non-covalent interactions, particularly hydrogen bonding, with the material surface) play a crucial role in forming a robust interface through in-situ polymerization. Achieving good macroscopic adhesion effects is contingent upon establishing a strong interface and augmenting the cohesive energy of the polymer. In essence, the concept of preparing highly adhesive materials through in-situ polymerization is indeed viable.

- Our research also highlights significant advantages over commercial adhesives in terms of environmental pollution and resource waste stemming from organic solvents and non-recyclable adhesives. As shown in **Figure R4**, currently commercially available reactive adhesives are only a few water-based (*Structural adhesive joints in engineering. Springer Science & Business Media, 1997. <https://doi.org/10.1007/978-94-009-5616-2>; Handbook of adhesive technology. CRC Press, 2017. <https://doi.org/10.1201/9781315120942>). In our work, we demonstrate that the performance of PPIL surpasses that of current water-based adhesives or solvent-free adhesives, such as pressure-sensitive and hot-melt adhesives. Commercialized water-based adhesives typically include polyvinyl acetate (PVAc), polyacrylate, polychloroprene, polyurethane adhesives, and a few water-based epoxy or phenolic adhesives (*Adhesive bonding: materials, applications, and technology. John Wiley & Sons, 2009. <https://doi.org/10.1002/9783527623921>). Despite claims of being distinct adhesive products due to modifications with fillers or additives for varied applications, their effective components remain fundamentally similar. Among these, PVAc emulsion adhesives are well-known in the consumer and industrial sectors. By incorporating solvents, a low-viscosity state is achieved, facilitating wetting of the solid material's surface. Subsequently, a diffusion process occurs between the adhesive molecules of both coats during and after pressure application, firmly bonding the adherents. The adhesives attain their final strength once the solvent evaporates or migrates into the substrate. The bond strength can increase, often within hours, after applying pressure. Solidification occurs through a drying process after joining the materials. In certain applications, such as woodworking with PVAc adhesives, a significant portion of the adhesive drying takes place through water sorption by the substrate, forming mechanical interlocking (*J. Appl. Polym. Sci. 2004, 91, 3009*). Although the adhesion strength of certain water-based epoxy or phenolic adhesives can be comparable to or even exceed that of hot-melt adhesives, both types form cross-linking structures during the curing process, albeit through different mechanisms and chemistries. However, after the service life, these adhesives with irreversible cross-linking contribute substantially to environmental problems and resource waste, accumulating as non-recyclable waste in significant quantities (*ACS Sustain. Chem. Eng. 2020, 8, 10767*).**

In order to address the reviewer's concern, we conducted a comprehensive assessment of the adhesion strengths of several commercially available adhesives, including water-based adhesives of PVAc, pressure-sensitive adhesives of double-tape, hot-melt adhesives of poly(ethylene-co-vinyl acetate (EVA), polyamide (PA), and polyolefins (PO), as well as

reactive adhesives of cyanoacrylates and epoxy resin, polyurethane (PUR) and silicone, as well as the corresponding discussion was provided in the section of “Adhesion performance”. The adhesion strengths of all these adhesives on various substrates at room temperature are presented in **Figure R5**. Even though the adhesion strengths of cyanoacrylates and epoxy resin are comparable, PPIL exhibits higher adhesion strengths than most commercially available adhesives on most substrates. In addition, unlike the high temperature of 150 °C required to soften and melt PA and PO for adhesion, the hot pretreatment of PPIL can be carried out at 110 °C, offering a more feasible and energy-efficient option for the adhesion process in hot-melt applications.

Moreover, several commercial cyanoacrylates and epoxy resin adhesives may not necessarily include volatile organic solvents theoretically, but they come with their own challenges. For instance, alkyl α -cyanoacrylate monomers must be meticulously stabilized during manufacturing, storage, and transport to prevent premature polymerization before the intended use, given their rapid setting characteristics at room temperature in the presence of moisture or weak bases (*Cyanoacrylate adhesives: A critical review. Reviews of Adhesion and Adhesives, 2016, 4, 4*). Acidic gases like hydrogen fluoride, nitric oxide, sulfur dioxide, and sulfur trioxide are employed during the manufacturing process to prevent premature polymerization by stabilizing the cyanoacrylate monomer in both liquid and gaseous phases (*Handbook of adhesive technology. CRC press, 2017. <https://doi.org/10.1201/978131512094>*). Moreover, cyanoacrylate adhesives predominantly utilize ethyl cyanoacrylate as their main component, which is a volatile monomer. Some monomer molecules can escape into the air as the glue dries, contributing to the characteristic odor and presenting safety risks. Even in small amounts, the impact of these volatile molecules can be significant. Although cyanoacrylate glues are typically used in small quantities, the concentration of odorant molecules in that small volume can be relatively high, making the smell seem more potent than it is. During the curing process, ethyl cyanoacrylate molecules react with moisture in the air, forming long polymer chains that create a strong adhesive bond. This chemical reaction may release small amounts of odorous byproducts. Furthermore, some additives in cyanoacrylate glues, such as plasticizers, may contain volatile organic solvents. Esters like ethyl acetate or butyl acetate, commonly found in cyanoacrylate glues, are considered volatile organic solvents. Additionally, most cyanoacrylate adhesives exhibit issues of brittleness, poor resistance to impact loads and organic solvents, and are relatively expensive. Epoxy adhesives, even though liquid precursors can be utilized, also contain various additional volatile organic solvents like wetting agents, rheology modifiers, adhesion promoters, and plasticizers (*Epoxy Adhesive Formulations. McGraw Hill Professional, New York, 2005, <https://doi.org/10.1036/0071455442>*). As acknowledged, epoxy adhesives will release some volatile organic compounds, such as acetone, benzene, and formaldehyde, during curing. These substances are harmful to human health, and long-term exposure may cause headaches, eye pain, respiratory tract irritation, and other

symptoms.

In order to address the reviewer’s concern, the corresponding results are provided in the revised Supplementary Information (Supplementary Fig. 20, and Supplementary Tables 2 - 5). The corresponding description in the revised manuscript was modified as follows: “*Lap shear tests were performed to quantitatively assess the adhesion strengths, yielding average adhesion strengths under ambient conditions on ceramic, epoxy (EP), and stainless steel (SS) of 15.6, 12.5, and 9.2 MPa, respectively (Fig. 3b), which are much higher than most available adhesives (Supplementary Fig. 20, Supplementary Tables 2 - 5)*”.

Figure R4. Summary of liquid-based adhesive types that have been reported so far

Figure R5. Adhesion strengths of commercially available adhesives.

3. Why 110 °C is chosen as the curing temperature throughout the test? How about other temperatures between 25 ~ 110 °C? If such high temperature is strictly required, this adhesive could only be applied in industrial scenarios. In fact, plenty of industrial in-situ adhesives present a shear strength of over 10 MPa.

Answer: Thank you for this comment.

The curing process in this study encompasses three key aspects: the removal of solvent, polymerization of monomers, and the formation of interfacial adhesion regions.

(1) Removal of Solvent:

- The solvent evaporation rate of water molecule increases with higher temperatures, significantly reducing the corresponding treatment time. However, excessively high treatment temperatures or heating rates can lead to rapid solvent evaporation, causing the formation of bubbles on the adhesive material surface and affecting the in-situ formation of the interface, as well as the process of achieving excellent interface contact.
- Maintaining a relatively high treatment temperature helps prevent solvent residue. Excessive water molecules can exert a strong plasticizing effect, weakening the interaction between polymer molecular chains. This is unfavorable for obtaining high-strength adhesion properties.

(2) Polymerization of Monomers:

- Higher temperatures enhance the movement ability of monomer molecules, effectively promoting the kinetic process of polymer reactions and allowing for rapid and high-level polymerization. Opting for a higher treatment temperature before reaching the thermal decomposition temperature is advantageous for this polymerization process.

(3) Formation of Interfacial Adhesion Regions:

- Elevated treatment temperatures contribute to increased mobility of polymer molecular chains. Above the glass transition temperature (T_g), higher temperatures enhance fluidity, enabling polymer chains to infiltrate the interface and fostering strong interfacial adhesion. The increase

in treatment temperature facilitates polymer chains' infiltration into the interface, promoting robust interfacial adhesion.

- Generally, lower viscosity formulations at high temperatures exhibit faster bonding (2 ~ 5 times) than higher viscosity formulations, significantly impacting gap-filling properties. This difference in viscosity influences bond strength flexibility and impact properties.

Taken together, the higher the curing temperature within the safe range, the more suitable for manual operation. Its selection is mainly determined by the fact that this temperature needs to be much greater than the T_g . Under temperature operating conditions with good fluidity, it will be more conducive to the inter-diffusion and interweaving of polymer chains and more conducive to the infiltration of the substrate surface. It should be pointed out here that excessively high processing temperatures may cause safety hazards to plant operators using equipment at such high temperatures. According to the reviewer's comment, we carefully compared our curing temperature and adhesion strength with reported adhesives, and these data are listed in **Table R3**. As observed, the curing temperature of PPIL (110 °C) aligns with most reported values and is generally considered safe for operation within a specific temperature range (80 ~ 110 °C). In contrast, as highlighted by Suresh S. Narine et al. (*Ind. Eng. Chem. Res.* 2008, 47, 7524), the application temperature of most commercial hot-melt adhesives in the industry typically needs to reach the average operating temperature of 170 ~ 180 °C.

- We would like to clarify that 110 °C is not necessarily the exclusive curing temperature selected during the test. In order to address the reviewer's concern, we conducted adhesion strength measurements at various curing temperatures ranging from 50 to 100 °C. As shown in **Figure R6**, even at low curing temperatures of 50 ~ 80 °C, conversion rates of 100% of monomer in different conditions were observed (**Figure R7**). Following isothermal treatments, PPIL exhibits a high adhesion strength ranging from 8.3 to 14.0 MPa (**Figure R8**). The overall adhesion performance at these temperatures remains higher than that of many adhesives, establishing its effectiveness at an acceptable temperature range.

In order to address the reviewer's concern, the corresponding results are provided in the revised Supplementary Information (Supplementary Figs. 25 - 27). The corresponding description in the revised manuscript was modified as follows: *“To address this concern, adhesion strength measurements were conducted at safer curing temperatures between 50 and 100 °C. As shown in Supplementary Figs. 25 and 26, even at low curing temperatures (50 to 80 °C), conversion rates of 100% of monomer were achieved. Following isothermal treatments, PPIL still exhibited a high adhesion strength ranging from 8.3 to 14.0 MPa (Supplementary Fig. 27)”*.

Table R3. Comparison of adhesion strength and curing temperature of PPILs and other reported functional adhesives.

Adhesive	Curing temperature (°C)	Adhesion strength (MPa)	Reference
P(VA-g-HBA)	140	17.3	Macromol. Rapid Commun. 2016, 37, 545
LC	100	1.6	Nat. Commun. 2016, 7, 12094
PDMS-COO-Zn	70	6.8	Nat. Commun. 2018, 9, 2725
BSA	95	4.0	J. Am. Chem. Soc. 2019, 141, 1359
DPU-HMA	130	5.7	Mater. Chem. Front. 2019, 3, 1833
2e+MX-154	205	20.5	Angew. Chem. Int. Ed. 2019, 58, 12271
SP-H-SB1%	120	2.0	Green Chem. 2020, 22, 1319
SEA0.2	80	10.2	ACS Materials Lett. 2021, 3, 7, 1003
IC-1	100	5.8	Angew. Chem. Int. Ed. 2021, 60, 8948
SiNP S-Bpin	215	39.6	Sci. Adv. 2021, 7, eabk2451
PTBN6	90	4.2	Adv. Funct. Mater. 2022, 2201959
BSA0.35	100	14.6	Adv. Sci. 2022, 9, 2203182
PGA-3	120	7.0	Adv. Mater. 2023, 35, 2300802
Soy-mal-tan	180	15.0	Nature 2023, 621, 306
A2	110	16.2	This work

Figure R6. Isothermal TGA of A2 monomer solution (concentration: 1 M) at different temperatures.

Figure R7. ^1H NMR spectrum (400 MHz, 25 °C, D_2O) of A2 monomer solution with a concentration of 1 M after different temperatures in isothermal curing treatments.

Figure R8. Adhesion strengths of A2 obtained by spontaneous polymerization with different temperatures in isothermal curing treatments.

4. In Figure 3e (SS substrate), the 2nd and 4th adhesion strength looks like higher than the original one. Such interesting results can be explained in the revised manuscript.

Answer: Thank you for this comment. It is essential to highlight that, despite achieving effective adhesive infiltration at the interface through our proposed in-situ spontaneous polymerization process, there is significant room for improvement at the microscopic scale. Challenges arise from

the presence of air on the surfaces of rough substrates, hindering complete wetting during the adhesion process (*Structural Adhesive Joints: Design, Analysis, and Testing*. John Wiley & Sons, 2020. ISBN: 978-1-119-73643-1). In repeated adhesion processes, the initially adhered parts of the substrate surface remain effectively adhered even after multiple cycles. However, parts that were not wetted during the initial process may become wetted, resulting in a slight improvement in the observed adhesion strength (**Figure R9**). It is noteworthy that the phenomenon of adhesion performance not decreasing but increasing after a certain number of cycles during cyclic adhesion processes has been reported multiple times in previous literature (*Adv. Funct. Mater.* 2018, 28, 1800848; *Sci. Adv.* 2018, 4, 8192; *ACS Appl. Mater. Interfaces* 2021, 13, 44860; *ACS Appl. Mater. Interfaces* 2022, 14, 27476; *ACS Materials Lett.* 2023, 5, 2528). Besides, it must also be admitted that this result may be related to certain experimental errors.

Figure R9. Schematic of proposed evolution of interface region during repeated cycling adhesion tests.

5. In the recipe of this adhesive material, AMPS and hydroxylamine are hydrophilic. Are all the samples in this work tested in a dry state? Or how about the stability of these adhesives under a high-humidity condition?

Answer: Thank you for this comment. All the samples in this work were tested in a dry state. We apologize for any confusion caused by the lack of a detailed description of our experimental procedures. Following the preparation of adhesively bonded joints through in-situ polymerization, we allow them to be set at room temperature with a humidity level of 30 ~ 40%RH for 3 h before conducting an adhesion performance test. However, due to the hydrophilicity of AMPS and hydroxylamine, the stability of most PPIL adhesives is unsatisfactory under high-humidity conditions (**Figure R10**). In order to address the reviewer's concern, we have proposed three different methods to alleviate this problem, considering physical blending modification, the chemical structure design of PPIL, and the coating design of the PPIL-adhered joints.

- Physical blending modification

PTFE is highly hydrophobic, and the hydrophilic/hydrophobic properties of the resulting composite after physical mixing with the polymer can be adjusted within a specific range. To leverage this property, we thoroughly mixed PTFE with the monomer aqueous solution before the in-situ spontaneous process. Subsequently, we isothermally treated it on the substrate surface to obtain a composite adhesive with reduced water absorption and improved stability in humid environments (**Figures R11 and R12**). As shown in **Figure R13**, after storing the

PPIL composite with 40 wt% PTFE in a 60 ~ 75%RH environment for 72 h, an adhesion strength of 2.5 MPa was observed.

- The chemical structure design of PPIL

As previously highlighted, the versatile in-situ spontaneous polymerization process we proposed for preparing adhesive materials can be applied to various structures. Various hydroxylamine molecules serve as raw materials to achieve in-situ polymerization, leading to the preparation of high-performance adhesive materials. Among these hydroxylamine molecules, N, N-dimethylethanolamine (for P6) is particularly advantageous for constructing PPIL with stability in high-humidity conditions. As shown in **Figure R14**, after storing the P6 in a 60 ~ 75%RH environment for 72 h, an adhesion strength of 5.7 MPa was observed.

- Coating design of the PPIL-adhered joints

Petrolatum, utilizing long-chain alkanes as a byproduct of crude oil refining as its active ingredient, is a white, uniform, odorless ointment known for its high hydrophobicity. We implemented coating modification based on the adhesive obtained after the in-situ polymerization of the monomer aqueous solution on the substrate surface. By applying a thin layer of petrolatum at room temperature to the exposed areas of the PPIL in the gaps of the bonding area, we could effectively create a waterproof membrane with excellent performance around the applied region. This membrane effectively blocks the diffusion of moisture in the air, which may lead to adhesion failure at different humidity levels (**Figure R15**). As shown in **Figure R16**, after storing in a 60 ~ 75%RH environment for 72 h, an adhesion strength of 15.5 MPa was observed (original value: 15.6 MPa). Even if the humidity is further increased to 75 ~ 90%RH, almost no significant reduction in adhesion properties was observed during this process.

In addition, the humidity-dependent adhesion may be advantageous for humidity sensing applications after functional modification, particularly for strong adhesion at room temperature (*Adv. Mater.* 2021, 2103674).

In order to address the reviewer's concern, the corresponding results are provided in the revised Supplementary Information (Supplementary Figs. 33 - 41). The corresponding description was added to the revised manuscript as follows: "*It should be noted that the adhesion strength of A2 adhesive may decline in humid conditions due to its water-solubility (Supplementary Fig. 33). However, methods like physical blending with hydrophobic PTFE, chemical structure design of PPIL, and coating design of PPIL-adhered joints can be applied to improve its stability in humid environments (Supplementary Figs. 34 - 41)*", and "*After creating adhesively bonded joints through in-situ polymerization, we let them cure at room temperature (30 ~ 40% RH) for 3 hours before assessing their adhesion performance*".

Figure R10. Adhesion strengths of A2 after storing in environmental conditions with different humidity.

PPII/PTFE composite (Reflection mode)

Figure R11. Light microscope (reflection mode) images of the PTFE and PPIL/PTFE.

Figure R12. SEM images of the PTFE and PPIL/PTFE.

Figure R13. Adhesion strengths of PPIL/PTFE composites after storing in environmental

conditions with different humidity.

Figure R14. Adhesion strengths of PPIL of P6 after storing in environmental conditions with different humidity.

Figure R15. SEM image of the interface between PPIL and the ceramic substrate with petrolatum coating.

Figure R16. Adhesion strengths of PPIL with petrolatum coating after storing in environmental conditions with different humidity.

Reviewer #2 (Remarks to the Author):

The work reports on a strategy to overcome the balance between cohesion and interfacial adhesion strength in adhesives. The strategy is based on the spontaneous polymerization of a protic ionic liquid (IL)-based monomer obtained through the neutralization of 2-acrylamide-2-methyl propane sulfonic acid and hydroxylamine. An initiator-free polymerization process is presented leading to the in-situ formation of a tough and thin adhesive layer with a highly entangled polymeric network and a strong interface contact between the adhesive and the substrate. The method also allows the development of adhesive composites with electrical conductivity or sensing functionality by incorporating specific fillers. The work is an area of strong and current interest, and in fact, a novel strategy for the development of high-performance IL-based adhesive is presented. The characteristics of the adhesive are convincingly demonstrated and the science behind the performance is properly discussed, including computer simulation. The work provides a mostly suitable description of the experimental work, allowing reproducibility in most cases. Most of the experimental data are properly discussed.

Response: We thank the reviewer for the positive evaluations of our work and constructive comments. The manuscript has been modified according to the reviewer's comments, and the changes are highlighted with a yellow background.

Detailed responses are listed as follows.

1. Adhesion assays were not performed following the standards (international norms) in the area. The authors should discuss why and how this affects potential applicability.

Answer: Thank you for this comment. In order to address the reviewer's concern, we provide the following response.

- The influence of the bonding area on adhesion strength during bonding is primarily manifested in the stress concentration phenomenon at the edge. As the bonding area increases, the edge stress concentration phenomenon becomes more pronounced, leading to a non-linear increase in observed adhesion force, i.e., a decrease in adhesion strength (*Goss B. Practical guide to adhesive bonding of small engineering plastic and rubber parts. ISmithers, 2010. ISBN: 978-1-84735-140-1; Adhesive Bonding Technology and Testing. John Wiley & Sons, 2023. ISBN: 978-3-527-83800-4*). By increasing the joint overlap, there is no significant change in the strength of the bond (**Figure R17**). This is because the joint starts to break at the stress peak at the end of the overlap, where the adhesion or cohesive strength of the adhesive is exceeded. By increasing the width of the joint, the shear stress distribution is not changed, so the failure load of lap joints increases in the same proportion as the joint width increases. Failure is most likely to occur at the ends where maximum stress is present.
- Despite acknowledging that the difference in bonding area can influence adhesion performance values, the strong adhesion of our glue poses a challenge during the actual test process. Conducting the adhesion test according to the standard bonding area consistently results in substrate failure (**Figure R18**). Consequently, valid adhesion performance data cannot be obtained. In order to address the reviewer’s concern, we also carefully compared the adhesion area used in this research with the reported literature, and these data are listed in **Table R4**. The adhesion area used in this research is consistent with most reported literature. From this perspective, it is reasonable to compare the adhesion performances of PPIL in this research with the values reported in the literature.

Figure R17. Schematic of the joint strength does not increase linearly with joint overlap due to the stress concentrations at the ends of the joint.

Figure R18. Photographs of the substrate failure during testing.

Table R4. Comparison of adhesion strength and adhesion area of PPILs and other reported functional adhesives.

Adhesive	Substrate	Area (mm * mm)	Adhesion strength (MPa)	Reference
2e+MX-154	Al	10 * 10	20.5	Angew. Chem. Int. Ed. 2019, 58, 12271
(A) _n	Glass	2.5 * 2.5	4.2	J. Am. Chem. Soc. 2020, 142, 2579
IC-1	Glass	5 * 5	5.8	Angew. Chem. Int. Ed. 2021, 60, 8948
pIG2-N0.4	Glass	15 * 10	4.7	Mater. Horiz. 2021, 8, 2057
SiNP S-Bpin	Glass	3 * 3	39.6	Sci. Adv. 2021, 7, eabk2451
P4-AS-PSS	Ceramic	20 * 15	1.2	Adv. Funct. Mater. 2021, 2109144
BSA0.35	SS	25 * 12.5	14.6	Adv. Sci. 2022, 9, 2203182
PVA/BA	Glass	10 * 10	0.6	PNAS 2022, 119, e2203074119
BN-6	SS	10 * 4	4.2	Adv. Funct. Mater. 2022, 2201959

ADM-4CTAB	SS	10 * 20	3.1	Angew. Chem. Int. Ed. 2022, e202204611
3I	SS	10 * 10	0.5	Adv. Mater. 2023, 35, 2208413
BSA	Titanium	10 * 10	4.0	Nat. Commun. 2023, 14, 5145
TFMD-2	Glass	10 * 10	1.2	Adv. Funct. Mater. 2023, 2304653
Soy-mal-tan	Al	12 * 12	15.0	Nature 2023, 621, 306
A2	Ceramic	10 * 10	16.2	This work

2. The authors must provide adhesion performance parameters comparison with related compounds form the literature.

Answer: Thank you for this comment. According to the reviewer's suggestion, we conducted a thorough comparison of our results with reported adhesives, and the corresponding data are detailed in **Table R1 - R3**. It is evident that our PPILs prepared through in-situ spontaneous polymerization of aqueous solution exhibit significant advantages in terms of adhesion strength at room temperature, adhesion strength at low temperatures, solvent resistance, and curing temperature. Additionally, our environmentally friendly preparation process employs cost-effective precursors (approximately \$0.032 per gram for raw materials) and eliminates organic solvent emissions. This not only contributes to a green synthesis but also allows for easy scalability, enabling the production of kilogram-scale highly adhesive poly(ionic liquid)s (**Supplementary Fig. 18**).

In order to address the reviewer's concern, the corresponding results are provided in the revised Supplementary Information (Supplementary Fig. 20, and Supplementary Tables 2 - 5). The corresponding description in the revised manuscript was modified as follows: "*Lap shear tests were performed to quantitatively assess the adhesion strengths, yielding average adhesion strengths under ambient conditions on ceramic, epoxy (EP), and stainless steel (SS) of 15.6, 12.5, and 9.2 MPa, respectively (Fig. 3b), which are much higher than most available adhesives (Supplementary Fig. 20, Supplementary Tables 2 - 5)*".

Table R1. Comparison of adhesion strength of PPILs and other reported functional adhesives at low temperatures.

Adhesive	Temperature (°C)	Adhesion strength (MPa)	Reference
----------	------------------	-------------------------	-----------

PC10-W1	-196	1.2	J. Am. Chem. Soc. 2020, 142, 21522
TA-epoxy	-196	0.6	Biomacromolecules 2022, 23, 3493
CA/PEG2000	-196	0.9	ACS Appl. Polym. Mater. 2022, 4, 4319
Poly(PA-H)	-60	0.4	ACS Appl. Mater. Interfaces 2022, 14, 27476
C12TAB/ChCl-urea	-80	1.0	Mater. Horiz. 2022, 9, 1700
BSA0.35	-196	9.5	Adv. Sci. 2022, 220318
P1	-196	4.8	ACS Nano 2022, 16, 5303
NIPA5.0	-196	15.9	Chem. Mater. 2023, 35, 7730
6-HTPB	-80	2.1	Green Chem. 2023, 25, 6845
PEA	-196	5.6	Eur. Polym. J. 2023, 198, 112387
Poly(A-C)	-196	9.5	Chem. Eng. J. 2023, 451, 138674
Poly(TA-DB)	-65	7.9	Chem. Eng. Sci. 2023, 281, 119164
cPA2	-196	17.4	J. Mater. Chem. A 2023, 11, 6286
DPETI	-196	2.2	Adv. Mater. 2023, 2310779
A2	-196	10.0	This work

Table R2. Comparison of organic solvent resistance of PPILs and other reported functional adhesives.

Adhesive	Types of organic solvents	Soaking time	Adhesion strength after soaking(MPa)	Reference
VPTA	2	22 d	N/A	J. Mater. Chem. A 2017, 5, 21169

DPU-HMA	1	18 h	2.7	Mater. Chem. Front. 2019, 3, 1833
CT-2	5	21 d	2.7	CCS Chem. 2020, 2, 1690
SP-DN	7	24 h	0.8	Mater. Horiz. 2021, 8, 2520
P4-AS-PAA	2	6 m	0.9	Adv. Funct. Mater. 2021, 2109144
HPU-HMA	3	30 d	N/A	Ind. Eng. Chem. Res. 2021, 60, 6925
P1	9	100 d	4.8	ACS Nano 2022, 16, 5303
C12TAB/ChCl-urea	4	N/A	0.6	Mater. Horiz. 2022, 9, 1700
BSA0.35	3	24 h	14.6	Adv. Sci. 2022, 9, 2203182
PVA-PTA	2	30 min	N/A	Adv. Funct. Mater. 2022, 2111892
PEG-TA	2	7 d	4.0	Macromol. Rapid Commun. 2022, 43, 2100830
NIPA5.0	9	168 h	18.8	Chem. Mater. 2023, 35, 7730
6-HTPB	4	4 h	1.1	Green Chem. 2023, 25, 6845
A2	8	1 m	12.8	This work

Table R3. Comparison of adhesion strength and curing temperature of PPILs and other reported functional adhesives.

Adhesive	Curing temperature (°C)	Adhesion strength (MPa)	Reference
P(VA-g-HBA)	140	17.3	Macromol. Rapid Commun. 2016, 37, 545
LC	100	1.6	Nat. Commun. 2016, 7, 12094

PDMS-COO-Zn	70	6.8	Nat. Commun. 2018, 9, 2725
BSA	95	4.0	J. Am. Chem. Soc. 2019, 141, 1359
DPU-HMA	130	5.7	Mater. Chem. Front. 2019, 3, 1833
2e+MX-154	205	20.5	Angew. Chem. Int. Ed. 2019, 58, 12271
SP-H-SB1%	120	2.0	Green Chem. 2020, 22, 1319
SEA0.2	80	10.2	ACS Materials Lett. 2021, 3, 7, 1003
IC-1	100	5.8	Angew. Chem. Int. Ed. 2021, 60, 8948
SiNP S-Bpin	215	39.6	Sci. Adv. 2021, 7, eabk2451
PTBN6	90	4.2	Adv. Funct. Mater. 2022, 2201959
BSA0.35	100	14.6	Adv. Sci. 2022, 9, 2203182
PGA-3	120	7.0	Adv. Mater. 2023, 35, 2300802
Soy-maltan	180	15.0	Nature 2023, 621, 306
A2	110	16.2	This work

3. The development of composites, both conductive and thermochromics is incomplete: important experimental information is missing, not allowing reproducibility and the characterization is incomplete in terms of morphological variations, filler dispersion and functional characteristics. This part must be extensively rewritten.

Answer: Thank you for pointing this out. According to the reviewer's suggestion, detailed experimental information was added to the revised manuscript and supplementary information. We also conducted characterization analyses of PPILs/MWCNTs and PPILs/thermo-chromic materials composites. As shown in **Figures R19, R20, R23, and R24**, both light microscopy and scanning electron microscopy (SEM) images revealed the homogeneous distribution of MWCNTs and thermo-chromic materials in the adhesive matrix. The fuzzy interface between the filler (MWCNTs and thermo-chromic materials) and the matrix material confirms the intimate microscopic interfacial

contact between the filler and the PPIL. For PPILs/MWCNTs composites, well-preserved characteristic peaks (D and G) of MWCNTs indicated that the simple preparation process of PPILs/MWCNTs has little influence on MWCNTs (**Figure R21**). Thus, the electronic conductivity of MWCNTs in the composite can be ensured. After incorporating MWCNTs or thermo-chromic materials into PPILs, the thermal stability of the material is almost not affected (**Figures R22 and R25**). PPILs, PPILs/MWCNTs, and PPILs/thermo-chromic materials exhibited the decomposition temperature of 254.3, 255.0, and 252.2 °C, respectively. Additionally, the viscoelastic behaviors of PPILs/thermo-chromic materials (A2 composite with a content of 20 wt% thermo-chromic materials) were investigated by temperature-dependent rheological measurements. As shown in **Figure R26**, the temperature-sensitive viscoelastic features of PPILs were preserved in the composite adhesives, exhibiting high reversibility in a temperature range of 10 ~ 110 °C. In addition, we evaluated the cyclic color transition performance and corresponding stability after prolonged exposure to different temperature conditions (–20 and 80 °C). As expected, even for a specific composite adhesive, the apparent color transition of the composite was observed after being continuously processed (**Figure R27**).

In order to address the reviewer's concern, the corresponding results are provided in the revised Supplementary Information (Supplementary Figs. 42 - 52). The corresponding description in the revised manuscript was modified as follows: "*Microscopic images, including light microscopy and scanning electron microscopy (Supplementary Figs. 42 and 43), illustrate the uniform distribution of TC materials in the adhesive matrix. The fuzzy interface between the filler and the matrix material validates the intimate microscopic interfacial contact between the filler and PPIL*", "*The variable temperature rheological test showed highly reversible viscoelastic behavior in the PPIL/TC composite. Additionally, the PPIL/TC composite displays stable and reversible color changes even after exposure to different temperatures for a week or undergoing ten cycles (Supplementary Figs. 44 - 46)*", "*For PPILs/MWCNTs composites, well-preserved characteristic peaks (D and G) of MWCNTs indicated that the simple preparation process of PPILs/MWCNTs has little influence on MWCNTs, endowing the composite adhesive with a high electrical conductivity of 1.53 S cm^{-1} (Supplementary Figs. 48 and 49). The in-situ spontaneously polymerized PPIL/MWCNT adhesive not only maintained the pristine thermal stability but also exhibited advantages in both adhesion strength and electrical conductivity compared to a reference composite fabricated using pre-polymerized A2 (12.7 MPa , 0.13 S cm^{-1} , Supplementary Figs. 49 and 50). Accordingly, the excellent electrical conductivity was attributed to an optimized configuration of molecular chains and conductive pathways^{40,41} (Supplementary Figs. 51 and 52)*", "*Typically, to prepare A2 composites with 20 wt% thermo-chromic materials, 3.12 g of spontaneously polymerized A2 was dissolved in 10 mL deionized water and mixed with 0.78 g thermo-chromic materials through 15 minutes of grinding. After 12 hours of mechanical stirring, the solvent was evaporated at 110 °C on the substrate surface. Following thermal treatment at 110 °C for 30 minutes, another pre-coated substrate was hot-pressed. This process yielded a thin adhesive layer (10 mm × 10 mm) with a*

thickness of approximately 50 μm , firmly adhering two substrates. Subsequently, after creating adhesively bonded joints through in-situ polymerization, they were allowed to set at room temperature (30 ~ 40% RH) for 3 hours before conducting an adhesion performance test”, and “Typically, to prepare A2 composites with 10 wt% inorganic or PTFE fillers, 0.35 g of functional fillers, such as MWCNTs, was dispersed in 10 ml A2ms (concentration: 1 M) by grinding for 15 minutes. After 12 hours of mixing and mechanical stirring, in-situ isothermal curing treatment (temperature: 110 $^{\circ}\text{C}$, time: 600 min) occurred on the substrate surface. Subsequently, another substrate was hot-pressed, resulting in firm adhesion with a 25 μm adhesive layer. Electronic conductivity measurements of the A2/MWCNTs composite adhesive layer were conducted using a standard four-probe method on an RTS-8 four-probe instrument, with a reference composite fabricated using pre-spontaneously polymerized A2 for comparison. That is, 3.12g of spontaneously polymerized A2 was dissolved in 10 ml deionized water and subsequently mixed with 0.35 g of MWCNTs fillers. For A2 composites with PTFE fillers, the hydrophobic nature necessitated over 40 minutes of grinding before mechanical stirring during preparation”.

Figure R19. Light microscope (including reflection and transmission mode) images of the MWCNTs and PPIL/MWCNTs.

Figure R20. SEM images of the MWCNTs and PPIL/MWCNTs.

Figure R21. Raman spectra of the MWCNTs and PPIL/MWCNTs.

Figure R22. TGA curves of PPIL/MWCNTs (T_d is defined as the decomposition temperature when 5% weight loss).

Figure R23. Light microscope (including reflection and transmission mode) images of the thermo-chromic materials and PPIL/thermo-chromic materials.

Figure R24. SEM images of the thermo-chromic materials and PPIL/thermo-chromic materials.

Figure R25. TGA curves. (a) The thermo-chromic materials, (b) PPIL/thermo-chromic materials (T_d is defined as the decomposition temperature when 5% weight loss).

Figure R26. Reversible temperature-dependent rheological test of the PPIL/thermo-chromic materials (angular frequency: 10 rad s^{-1} , strain: 1%).

Figure R27. Macroscopic stability test of PPIL/thermo-chromic materials after storing in environmental conditions with different temperatures.

4. There are no error bars and uncertainties in several parts on the work, event when the experimental variations are often within experimental errors.

Answer: Thank you for pointing this out. Error bars have been added to the figures in both the revised manuscript and supplementary information for all tests.

Reviewer #3 (Remarks to the Author):

The contribution from Zhang et al on polymerized adhesives is interesting. I believe the study is well performed and thorough. The results are novel, though I am unsure there is a large advance over state of the art here. The use of ionic polymers in adhesives is not unknown, and though these are referred to as protic ionic liquids, the classification is dubious (they are fairly standard low molecular weight ionic polymers).

Response: We thank the reviewer for the positive evaluations of our work and constructive comments. The manuscript has been modified according to the reviewer's comments, and the changes are highlighted with a yellow background.

At first, it is crucial to clarify that the poly(ionic liquid) adhesives discussed in our work denote polymer materials formed through an in-situ spontaneous polymerization process and do not pertain to the monomer aqueous solution. As outlined in the initial supplementary information, gel permeation chromatography was employed to determine the molecular weights (M_w) and polydispersity (PDI) of protic poly(ionic liquid)s. The analysis confirmed average M_w and PDI values exceeding 100 kDa and 3, respectively (**Supplementary Table 1**). This outcome indicates that the protic poly(ionic liquid)s consists, on average, of 343 repeating monomers per chain, aligning with the polymer or macromolecule definition set by the International Union of Pure and Applied Chemistry standards (*Pure Appl. Chem.* 1996, 68, 2287).

The polymer obtained through in-situ spontaneous polymerization in this study can be classified as protic poly(ionic liquid), aligning with prior literature based on the preparation method and structural composition (*ACS Appl. Mater. Interfaces* 2019, 11, 6111; *Green Chem.* 2021, 23, 9922; *Macromol. Chem. Phys.* 2022, 223, 2200124; *Encyclopedia of Ionic Liquids*. Springer Nature, 2022, <https://doi.org/10.1007/978-981-33-4221-7>). Meanwhile, we agree with the reviewer's comment that the protic poly(ionic liquid) can be defined as a sub-branch of the ionic polymers (or polyelectrolytes) (*Specialty polymers*. London: Blackie, 1987 ISBN: 978-1-4615-7894-9; *Developments in Ionic Polymers—2*. Springer Science & Business Media, 2012. <https://doi.org/10.1007/978-94-009-4187-8>). In this research, the polymerizable component is the 2-acrylamido-2-methyl-1-propanesulfonic acid (AMPS) monomer, a crystalline compound with a 192 °C melting point. It is converted to an ammonium salt by adding an equimolar quantity of hydroxylamine molecules. This amine was chosen for its ability to solvate the AMPS monomer, and its hydroxylamine substituents could shield the protonated ionic center that coordinates with the sulfonated anion, thereby effectively lowering the melting point or glass transition to a very low temperature ($T_g = 55$ °C). Considering that conventional ionic polymers are typically solid with a T_g far beyond room temperature, e.g., poly(sodium 2-acrylamido-2-methylpropanesulfonate), $T_g \approx 168$ °C, and poly(acrylic acid) sodium salt, $T_g \approx 230$ °C, we are inclined to categorize this material as poly(ionic liquid)s. However, if the reviewer insists that referring to it as ionic polymer is more appropriate, we are willing to alter the definition of this polymer from protic poly(ionic liquid) to ionic polymer.

Detailed responses are listed as follows.

1. What is the real advance here? I don't see what properties are much better than state-of-the-art. The authors need to clarify and highlight this.

Answer: Thank you for this comment. It seems that our manuscript did not adequately highlight the genuine advancement of our work. To address the reviewer's concern, we provide a detailed clarification as follows. Additionally, we meticulously compare the properties and performance of poly(ionic liquid) adhesives produced through facile in-situ spontaneous polymerization with state-of-the-art counterparts.

- Various efficient catechol- and non-catechol-based modern adhesives have been developed, while the persistent challenge of balancing the cohesion and interfacial adhesion strength remains a hindrance to further improving the overall adhesion performance. Conventional approaches to increasing the cohesion strength often lead to a substantial reduction in interfacial adhesion strength, thus ultimately deteriorating the resulting adhesion performance. Inspired by the natural adhesion in organisms such as mussels and sandcastle worms, **we herein created poly(ionic liquid) adhesives by enhancing structural design and employing in-situ spontaneous polymerization of readily available and cost-effective precursors in aqueous solutions in the absence of any other agents (initiators, catalysts, cross-linkers). These adhesives offer numerous advantages, including the absence of volatile organic compound emissions, environmental friendliness, cost-effectiveness, straightforward synthesis for large-scale production, high adhesion strength, outstanding performance at extremely low temperatures, exceptional resistance to organic solvents, and the capability to prepare multi-functional composite materials.**
- This is the first report that poly(ionic liquid) materials can be simply prepared by neutralization reaction and spontaneous polymerization. Almost 100% yield quantitative synthesis of poly(ionic liquid)s in this work could be advantageous for its various large-scale applications. Such a method of designing poly(ionic liquid) materials has never been achieved before, as most poly(ionic liquid)s are related to tedious synthesis and relatively low reaction yield, which seriously limit their practical application. Considering the widespread attention and promising application prospects of poly(ionic liquid) materials in various fields, including energy storage and conversion, we posit that the in-situ spontaneous polymerization for poly(ionic liquid) presented in our study can offer crucial inspiration and guidance for other applications.
- The neutralization reaction and spontaneous polymerization utilize cost-effective precursors (approximately \$0.032 \$ g⁻¹ for raw materials) and avoid organic solvent emissions, facilitating easy scalability for the preparation of **kilogram-scale** highly adhesive poly(ionic liquid)s (Large-scale experiments for industrial applications are ongoing in our team). This stands in contrast to the often intricate synthesis and low reaction yields associated with many poly(ionic liquid)s, which significantly hinder their practical utility. Achieving almost 100% yield in the

quantitative synthesis of poly(ionic liquid)s holds promise for a range of large-scale applications, including adhesion, coating, media, and electrolytes.

- The resulting poly(ionic liquid) exhibits durable, long-term, and recyclable adhesion on diverse substrates, including ceramic, epoxy, and stainless steel. Its maximum adhesion strength reaches **16.2 MPa**, surpassing that of most catechol and commercially available adhesives. Remarkably, even at the ultra-low temperature of liquid nitrogen ($-196\text{ }^{\circ}\text{C}$), adhesion strengths of 10.0, 5.0, and 6.8 MPa are maintained on ceramic, epoxy (EP), and stainless steel (SS) surfaces, respectively. In order to address the reviewer's concern, we meticulously compared the low-temperature adhesion strength of PPIL with reported adhesives, as detailed in **Table R1**. Notably, the performance of PPIL exceeds that of the majority of currently reported adhesive materials. Typically, the thermal expansion and mechanical properties of adhesives change with temperature, leading to uneven stress distribution and interface stress concentration in a single lap joint during temperature fluctuations. This can result in the fracture of brittle adhesive materials at low temperatures. This study reveals that in-situ polymerization effectively mitigates this issue. Ongoing research within our team focuses on achieving ultra-strong low-temperature adhesion through an in-situ curing process, with relevant results to be reported in subsequent research.
- Because of their **unique organic solvents-resistant ability**, the poly(ionic liquid)s-adhered joints were insoluble and stable in a variety of common organic solvents, such as n-hexane, dichloromethane, ethyl acetate, tetrahydrofuran, ethanol, acetone, isopropanol, and acetonitrile. According to the reviewer's suggestion, we carefully compared the organic solvent resistance of PPIL and other reported functional adhesives and these data are listed in **Table R2**.
- The adhesion of these poly(ionic liquid)s was found to be highly affected by their chemical structure, and the adhesive mechanism was carefully investigated by a combined experimental study and theoretical calculation. The in-situ constructed highly interconnected polymeric networks with abundant H-bonding and electrostatic interactions accounted for the improved and balanced cohesion and the interfacial interactions. Additionally, the in-situ spontaneous polymerization process we propose to prepare protic poly(ionic liquid) adhesive materials is highly versatile and can be used in many structures. Various hydroxylamine molecules can be used as raw materials to achieve in-situ polymerization to prepare high-performance adhesive materials.
- Visualized sensing functionality composite adhesives were further fabricated after the incorporation of fillers. For example, an obvious color transition (from deep blue to rose red) of the adhesive composite was observed upon heating from -20 to $80\text{ }^{\circ}\text{C}$, associated with adhesion strength increasing from 2.1 to 5.4 MPa. Non-destructive and continuous visual monitoring of the adhesion strength can be realized without high cost and tedious synthesis.
- The in-situ spontaneous polymerization process can be used to prepare functional composite materials with excellent comprehensive properties in other fields, such as application as a

binder component in the battery. To address the reviewer's concerns, we applied this process to PPIL. By uniformly mixing the aqueous monomer solution with carbon nanotubes and graphite and conducting in-situ polymerization on the current collector's surface, we produced a material that adheres robustly to the collector after prolonged immersion in the electrolyte within the battery. During performance testing, it exhibited superior cycle stability and rate performance compared to many reported materials (*Adv. Funct. Mater.* 2023, 33, 2302951; *ACS Energy Lett.* 2023, 8, 1336; *J. Power Sources* 2019, 429, 67; *ACS Sustainable Chem. Eng.* 2022, 10, 12023; *Chem. Eng. J* 2023, 476, 146299). Beyond serving as a binder for the graphite anode, it also performs well as a cathode binder for lithium iron phosphate. This application underscores the practical significance of in-situ polymerization across multiple fields.

In order to address the reviewer's concern, the corresponding results are provided in the revised Manuscript (Fig. 3f) and Supplementary Information (Supplementary Figs. 22 - 24, Supplementary Figs. 28 and 29, and Supplementary Figs. 53 - 59). The corresponding description in the revised manuscript was modified as follows: *"Nonetheless, the use of these adhesives in large quantities poses significant characteristic odors and environmental issues and results in wastage of resources due to their volatile organic solvents or irreversible adhesion process", "Increasing the temperature, as shown by employing a higher-temperature laboratory heat gun (around 170 °C) for 2 minutes, markedly speeds up the polymerization reaction kinetics, resulting in a 100% conversion rate and substantial adhesion strength of 10 MPa (Supplementary Figs. 22 - 24)", "In traditional single lap joints, challenges arise from uneven stress distribution and interface stress concentration caused by variations in thermal expansion and mechanical properties with temperature changes. This tendency often results in the brittleness of the adhesive material, especially at low temperatures. Notably, even at the extremely low temperature of liquid nitrogen (-196 °C), the A2 adhesive exhibited significant adhesion strengths of 10.0, 5.0, and 6.8 MPa on ceramic, EP, and SS surfaces, respectively", "The adhesion effect depends on the chemical structure. Various hydroxylamine molecules could serve as raw materials for in-situ polymerization, facilitating the preparation of high-performance adhesive materials (Fig. 3f, and Supplementary Figs. 28 and 29)", and "The highly conductive PPIL/MWCNT adhesive, serving as a binder/conductive additive composite component, demonstrates its utility in batteries. As an initial application, we employ it as a binder/conductive additive for a graphite anode in lithium-ion batteries. After uniformly mixing the aqueous monomer solution with MWCNTs and graphite, in-situ polymerization was conducted on the current collector surface. As demonstrated in Supplementary Figs. 53 and 54, the resulting material exhibits robust adhesion to the current collector, maintaining exceptional cycle stability and rate performance in battery tests, even after 250 and 400 charge-discharge cycles at 1 C and 2 C rates, with discharge capacities of approximately 354.2 and 306.6 mAh g⁻¹, respectively (Fig. 4i, and Supplementary Figs. 55 and 56). Additionally, the PPIL/MWCNT adhesive showcases excellent performance as a cathode binder for lithium iron phosphate (LiFePO₄) (Supplementary Figs. 57*

and 58). After 35 cycles of charge-discharge cycling at the 0.2 C rate, the discharge capacity obtained is approximately 147.0 mAh g⁻¹ (Supplementary Fig. 59)''.

Table R1. Comparison of adhesion strength of PPILs and other reported functional adhesives at low temperatures.

Adhesive	Temperature (°C)	Adhesion strength (MPa)	Reference
PC10-W1	-196	1.2	J. Am. Chem. Soc. 2020, 142, 21522
TA-epoxy	-196	0.6	Biomacromolecules 2022, 23, 3493
CA/PEG2000	-196	0.9	ACS Appl. Polym. Mater. 2022, 4, 4319
Poly(PA-H)	-60	0.4	ACS Appl. Mater. Interfaces 2022, 14, 27476
C12TAB/ChCl-urea	-80	1.0	Mater. Horiz. 2022, 9, 1700
BSA0.35	-196	9.5	Adv. Sci. 2022, 220318
P1	-196	4.8	ACS Nano 2022, 16, 5303
NIPA5.0	-196	15.9	Chem. Mater. 2023, 35, 7730
6-HTPB	-80	2.1	Green Chem. 2023, 25, 6845
PEA	-196	5.6	Eur. Polym. J. 2023, 198, 112387
Poly(A-C)	-196	9.5	Chem. Eng. J. 2023, 451, 138674
Poly(TA-DB)	-65	7.9	Chem. Eng. Sci. 2023, 281, 119164
cPA2	-196	17.4	J. Mater. Chem. A 2023, 11, 6286
DPETI	-196	2.2	Adv. Mater. 2023, 2310779
A2	-196	10.0	This work

Table R2. Comparison of organic solvent resistance of PPILs and other reported functional adhesives.

Adhesive	Types of organic solvents	Soaking time	Adhesion strength after soaking(MPa)	Reference
VPTA	2	22 d	N/A	J. Mater. Chem. A 2017, 5, 21169
DPU-HMA	1	18 h	2.7	Mater. Chem. Front. 2019, 3, 1833
CT-2	5	21 d	2.7	CCS Chem. 2020, 2, 1690
SP-DN	7	24 h	0.8	Mater. Horiz. 2021, 8, 2520
P4-AS-PAA	2	6 m	0.9	Adv. Funct. Mater. 2021, 2109144
HPU-HMA	3	30 d	N/A	Ind. Eng. Chem. Res. 2021, 60, 6925
P1	9	100 d	4.8	ACS Nano 2022, 16, 5303
C12TAB/ChCl-urea	4	N/A	0.6	Mater. Horiz. 2022, 9, 1700
BSA0.35	3	24 h	14.6	Adv. Sci. 2022, 9, 2203182
PVA-PTA	2	30 min	N/A	Adv. Funct. Mater. 2022, 2111892
PEG-TA	2	7 d	4.0	Macromol. Rapid Commun. 2022, 43, 2100830
NIPA5.0	9	168 h	18.8	Chem. Mater. 2023, 35, 7730
6-HTPB	4	4 h	1.1	Green Chem. 2023, 25, 6845
A2	8	1 m	12.8	This work

2. I don't think the lack of solvents is much of a sustainability advance, given the use of volatile organic amines in the production.

Answer: Thank you for this comment. In order to address the reviewer's concern, we provide the following response.

- Firstly, while hydroxylamine molecules, a key raw material in our synthesis, are commonly considered volatile organic compounds, it is crucial to note that the hydroxylamine compounds used in this study typically possess higher boiling points (T_b), often exceeding 200 °C (**Table R5**). The proposed in-situ spontaneous polymerization method for preparing highly adhesive protic poly(ionic liquid) materials is adaptable to various hydroxylamine molecules with diverse structures. The polymerization process demonstrates consistency across different hydroxylamine molecules, ensuring high conversion rates for their monomers (**Supplementary Fig. 28**). Consequently, not only volatile hydroxylamine molecules but also non-volatile ones can be effectively utilized as raw materials for producing highly adhesive protic poly(ionic liquid) materials. For instance, in addition to ethanolamine (for A1), diethanolamine (for A2), and triethanolamine (for A3) mentioned earlier, various hydroxylamine molecules, including those in a non-volatile solid state, are suitable for synthesis (**Table R5**, *diethylaminoethanol for P1, 2-(ethylamino)ethan-1-ol for P2, 3-amino-1,2-propanediol for P3, 2,2,2,2-ethylenedinitrilotetraethanol for P4, 2-(cyclohexylamino)ethanol for P5, N, N-dibenzylethanolamine for P6, tris(hydroxymethyl)aminomethane for P7, N-methyl-D-glucamine for P8, N-phenyldiethanolamine for P9, 8-aminooctan-1-ol for P10, 2-(phenylamino)ethanol for P11, 4-aminocyclohexan-1-ol for P12, 1-deoxy-1-(n-octylamino)-D-glucitol for P13, (4-(aminomethyl)phenyl)methanol for P14, (2S,3S,4R)-2-aminooctadecane-1,3,4-triol for P15*).
- Secondly, it is noteworthy that hydroxylamines are not directly used as volatile organic compounds in this work. Instead, they are initially mixed with AMPS in water and heated to achieve in-situ spontaneous polymerization, resulting in highly adhesive protic poly(ionic liquid) materials. In other words, during the synthetic procedure, organic amines undergo protonation at room temperature and are no longer in their original free state. This protonation significantly enhances the thermal stability of hydroxylamine compounds (*J. Am. Chem. Soc. 2014, 136, 1690; Chem. Mater. 2014, 26, 2915; Nano Energy 2015, 13, 376*), and our thermogravimetric analysis (TGA) on pure AMPS and diethanolamine (DEA) further confirmed this point. As shown in **Figure R28**, almost all DEA evaporated, while AMPS exhibited high stability with only a slight weight loss due to free water. In contrast, for the monomer solution containing AMPS and DEA, significant weight loss occurred only in the initial 20 minutes of the isothermal TGA test at 110 °C (**Figure 2b** in the revised manuscript). The remaining weight was consistently aligned with the solid content ratio in an aqueous solution at a theoretical concentration of 1 M A2 monomer (experimental weight ratio: 25.7%, theoretical weight ratio: 23.8%). Based on these observed experimental phenomena, we can affirm the assumption that there will be no emission of hydroxylamine molecules during the isothermal in-situ polymerization process.

Therefore, while some hydroxylamine molecules with low boiling points are considered volatile organic compounds, the majority of them possess higher boiling points, often exceeding 200 °C. Meanwhile, all hydroxylamines are not directly employed as volatile organic compounds in this study; instead, they function as reactants for protonation in water within a monomer solution for in-situ spontaneous polymerization. This protonation significantly enhances the thermal stability of hydroxylamine compounds. Consequently, both the synthesis and application of protic poly(ionic liquid) adhesives can effectively avoid the emission of volatile organic compounds.

In order to address the reviewer's concern, the corresponding results are provided in the revised Manuscript (Fig. 3f) and Supplementary Information (Supplementary Fig. 8, and Supplementary Figs. 28 and 29). The corresponding description in the revised manuscript was modified as follows: *“The adhesion effect depends on the chemical structure. Various hydroxylamine molecules could serve as raw materials for in-situ polymerization, facilitating the preparation of high-performance adhesive materials (Fig. 3f, and Supplementary Figs. 28 and 29)”*, and *“It should be noted that the neutralization reaction significantly inhibited the volatility and enhanced the thermal stability of the hydroxylamine molecule (Supplementary Fig. 8). The residual weight of A2ms consistently correlated with the solid content ratio in a 1 M theoretical concentration aqueous solution (experimental weight ratio: 25.7%, theoretical weight ratio: 23.8%)”*.

Table R5. Comparison of boiling points of the hydroxylamines used in this work.

Hydroxylamine	T_b (°C)
Ethanolamine (for A1)	170
Diethanolamine (for A2)	271.1
Triethanolamine (for A3)	335.4
Diethylaminoethanol (for P1)	161
2-(Ethylamino)ethan-1-ol (for P2)	167
3-Amino-1,2-propanediol (for P3)	264
2,2,2,2-Ethylenedinitrilotetraethanol (for P4)	327.2
2-(Cyclohexylamino)ethanol (for P5)	266.7
N, N-Dibenzylethanolamine (for P6)	206 °C (15mm Hg)
Tris(hydroxymethyl)aminomethane (for P7)	219
N-Methyl-D-glucamine (for P8)	490
N-Phenyldiethanolamine (for P9)	270
8-Aminoctan-1-ol (for P10)	232.9
2-(Phenylamino)ethanol (for P11)	286.1
4-Aminocyclohexan-1-ol (for P12)	201.1

1-Deoxy-1-(n-octylamino)-D-glucitol (for P13)	524.7
(4-(Aminomethyl)phenyl)methanol (for P14)	280.0
(2S,3S,4R)-2-Aminooctadecane-1,3,4-triol (for P15)	483.7

Figure R28. Isothermal TGA and TGA curves. (a) and (b) DEA, (c) and (d) AMPS (T_d is defined as the decomposition temperature when 5% weight loss).

3. What evidence is there for conductive applications? I see it as possible, but as yet unverified.

Answer: Thank you so much for this comment. According to the reviewer's suggestion, we utilize the conductive composite adhesive derived from in-situ spontaneously polymerized PPIL/MWCNT as a conductive binder in lithium-ion batteries (LIBs), resulting in outstanding battery performance. It is recognized that, despite their limited quantity, binders play a pivotal role in battery performance. Their essential function involves preserving the electrode's integrity by binding the active material and conductive additives together, in addition to connecting them to the current collector during the charge-discharge process.

In contrast to conventional anode/cathode assembly methods, which entail physically mixing active material, polymer binder, and conductive material, our approach involves initially blending active material, conductive material, and the precursor solution of PPILs. Subsequent in-situ spontaneous polymerization achieves both uniform dispersion of the active material and intimate

contact between the active material and conductive material, facilitated by the in-situ constructed binder. As a proof of concept, graphite was employed as the active material for LIBs, uniformly mixed with the A2 monomer solution containing MWCNTs and applied onto the current collector surface. Following in-situ polymerization to create the target anode, the battery was assembled using commercially available lithium foil, separator, and electrolyte. Examination of electrode morphology revealed an even mixture of graphite, MWCNT, and PPIL, forming close contact with the current collector (**Figures R29 and R30**). Consistent with our prior work on polymer binders in batteries (*Nano Materials Science* 2021, 3, 124; *J. Mater. Chem. A*, 2021, 9, 2375), this interconnected and porous surface structure of the electrodes enhances battery charge-discharge processes. As expected, after 250 and 400 charge-discharge cycles at rates of 1 C and 2 C, the discharge capacities were approximately 354.2 and 306.6 mAh g⁻¹, respectively (theoretical capacity: 372 mAh g⁻¹) (**Figures R31 and R32**). The exceptional cycle stability and rate performance affirm that in-situ constructed PPIL-based graphite anodes surpass those employing most commercial binders (*J. Power Sources* 2019, 429, 67; *ACS Sustainable Chem. Eng.* 2022, 10, 12023; *Chem. Eng. J* 2023, 476, 146299; *Adv. Funct. Mater.* 2023, 33, 2302951; *ACS Energy Lett.* 2023, 8, 1336). Beyond graphite anodes, the in-situ construction method for electrodes is also applicable to create cathode electrodes using lithium iron phosphate (**Figures R33-R35**). After 35 cycles of charge-discharge cycling at a 0.2 C rate, the discharge capacity was approximately 147.0 mAh g⁻¹ (theoretical capacity: 170 mAh g⁻¹).

Given their conductive and adhesive characteristics, along with the straightforward and in-situ construction method, these conductive composite adhesives hold promise for diverse applications in various fields, particularly in energy storage and conversion.

In order to address the reviewer's concern, the corresponding results are provided in the revised Supplementary Information (Supplementary Figs. 53 - 59). The corresponding description was added to the revised manuscript as follows: "*The highly conductive PPIL/MWCNT adhesive, serving as a binder/conductive additive composite component, demonstrates its utility in batteries. As an initial application, we employ it as a binder/conductive additive for a graphite anode in lithium-ion batteries. After uniformly mixing the aqueous monomer solution with MWCNTs and graphite, in-situ polymerization was conducted on the current collector surface. As demonstrated in Supplementary Figs. 53 and 54, the resulting material exhibits robust adhesion to the current collector, maintaining exceptional cycle stability and rate performance in battery tests, even after 250 and 400 charge-discharge cycles at 1 C and 2 C rates, with discharge capacities of approximately 354.2 and 306.6 mAh g⁻¹, respectively (Fig. 4i, and Supplementary Figs. 55 and 56). Additionally, the PPIL/MWCNT adhesive showcases excellent performance as a cathode binder for lithium iron phosphate (LiFePO₄) (Supplementary Figs. 57 and 58). After 35 cycles of charge-discharge cycling at the 0.2 C rate, the discharge capacity obtained is approximately 147.0 mAh g⁻¹ (Supplementary Fig. 59)*".

Figure R29. SEM images of the graphite, MWCNTs, graphite/MWCNTs, and graphite/PPIL/MWCNTs on Cu foil (the current collector).

Figure R30. ^1H NMR spectrum (400 MHz, 25 °C, $\text{DMSO-}d_6$) of commercially available graphite electrolytes before and after the anode is immersed for 2 weeks. Prolonged immersion of the graphite/PPIL/MWCNTs anode in the electrolyte did not cause any network disruption, and there were no observed NMR signals indicating the presence of polymeric adhesives.

Figure R31. Charge-discharge voltage profiles and cycle performance of the graphite/Li cells at current densities of 2 C. The inset demonstrates the ignition capability of LED lamps using the graphite/Li cell subjected to 400 charge-discharge cycles.

Figure R32. Cycle performance of the graphite/Li cells at current densities of 1 C.

Figure R33. SEM images of LFP, MWCNT, LFP/MWCNT, and LFP/PPIL/MWCNT on Al foil (the current collector).

LFP electrolyte before cathode being immersed

LFP electrolyte after cathode being immersed for 2 weeks

0 7.5 7.0 6.5 6.0 5.5 5.0 4.5 4.0 3.5 3.0 2.5 2.0 1.5 1.0 0.5

Figure R34. ^1H NMR spectrum (400 MHz, 25 °C, $\text{DMSO-}d_6$) of solvated ionic liquid, i.e., tetraglyme lithium bis(trifluoromethanesulfonyl)amide, electrolytes before and after the cathode is immersed for 2 weeks. Prolonged immersion of the LFP/PPIL/MWCNT cathode in electrolytes did not lead to network disruption, and no NMR signals associated with polymeric adhesives were detected.

Figure R35. Charge-discharge voltage profiles and cycle performance of the LFP/Li cells at current densities of 0.2 C.

REVIEWERS' COMMENTS

Reviewer #1 (Remarks to the Author):

The authors have addressed all issues. The paper can be accepted.

Reviewer #2 (Remarks to the Author):

The authors have performed a thorough revision of the manuscript, taking into consideration all comments from the reviewers. New experimental work has been performed and the discussion has been greatly improved, also comparing the results with the literature at a performance level.

This referee supports publication of the work in the present form.

Reviewer #3 (Remarks to the Author):

I think some of the concerns have been considered by the authors, but I am now much more sceptical of the cost here (their quoted prices are not accurate from my listings in the Chemical Price Index), which was a comment from Reviewer 1.

For my comments, I am still not sure what the real advance is here. The only novelty I can see is that they used an ionic polymer which is different from other ionic polymers for this application. It doesn't seem to be better than the state of the art, and the sustainability claims are more than dubious - alkanolamines are well known to be both toxic and corrosive. I agree one could call this a "poly ionic liquid" but that is irrelevant to the novelty.

The point-to-point response to the reviewers' comments

Reviewer #1 (Remarks to the Author):

The authors have addressed all issues. The paper can be accepted.

Response: We thank the reviewer for the very positive evaluation of our work.

Reviewer #2 (Remarks to the Author):

The authors have performed a thorough revision of the manuscript, taking into consideration all comments from the reviewers. New experimental work has been performed and the discussion has been greatly improved, also comparing the results with the literature at a performance level. This referee supports publication of the work in the present form.

Response: We thank the reviewer for the very positive evaluation of our work.

Reviewer #3 (Remarks to the Author):

I think some of the concerns have been considered by the authors, but I am now much more sceptical of the cost here (their quoted prices are not accurate from my listings in the Chemical Price Index), which was a comment from Reviewer 1.

For my comments, I am still not sure what the real advance is here. The only novelty I can see is that they used an ionic polymer which is different from other ionic polymers for this application. It doesn't seem to be better than the state of the art, and the sustainability claims are more than dubious - alkanolamines are well known to be both toxic and corrosive. I agree one could call this a "poly ionic liquid" but that is irrelevant to the novelty.

Response: We thank the reviewer for the comments.

Detailed responses are listed as follows:

- It is important to acknowledge that there may be significant price variations for the same experimental chemical in different regions. To confirm this, one can check real-time prices in different areas. As depicted in Figure R1, the current price of 2-acrylamide-2-methylpropanesulfonic acid (AMPS) on Sigma-Aldrich is 6255.14 CNY per 25 kg, which is equivalent to 0.035 USD g⁻¹ (the previously mentioned "0.029 USD g⁻¹" in the manuscript, exchange rate of 2024-03-29: 1 USD = 7.22 CNY). Similarly, the current price of 2,2-iminodiethanol (DEA) on Sigma-Aldrich is 5307.41 CNY per 50 L (as depicted in Figure R2), which

is equivalent to 0.015 USD mL⁻¹ (the previously mentioned "0.043 USD mL⁻¹" in the manuscript). In contrast, these chemicals are much cheaper in our local area (China). As depicted in Figure R3, the current price of AMPS is 1853 CNY per 25 kg, which is equivalent to 0.010 USD g⁻¹ (even lower than the previously mentioned "0.029 USD g⁻¹" in the manuscript). Similarly, the current price of DEA is 918 CNY per 25 L (as depicted in Figure R4), which is equivalent to 0.005 USD mL⁻¹ (also lower than the previously mentioned "0.043 USD mL⁻¹" in the manuscript). It should be noted that while there may be some variation in the cost of chemicals across regions and countries, the majority of the reagents utilized in this research are commercially manufactured in large quantities, resulting in relatively low costs. However, if the reviewer insists that the cost of our chemicals is inappropriate, we can change it according to the reviewer's request.

[REDACTED]

Figure R1. The current price of AMPS on Sigma-Aldrich.

https://www.sigmaaldrich.cn/CN/zh/search/15214-89-8?facet=facet_brand%3ASigma-Aldrich&focus=products&page=1&perpage=30&sort=relevance&term=15214-89-8&type=product

[REDACTED]

Figure R2. The current price of DEA on Sigma-Aldrich.

<https://www.sigmaaldrich.cn/CN/zh/substance/diethanolamine10514111422>

[REDACTED]

Figure R3. The current price of AMPS on Adamas (China).

https://www.tansoole.com/upload/detail/01/D5VB_7RDF_013343964.html?v=2024032321222224

236174313427

[REDACTED]

Figure R4. The current price of DEA on Adamas (China).

https://www.tansoole.com/upload/detail/01/0CQ3_YJ3J_01271548.html?v=2024032321222224236174313427

- Next, we would like to highlight the advancements of our work again, in order to address the reviewer's concern.
- (1) These adhesives offer numerous advantages, including the absence of volatile organic compound emissions, environmental friendliness, cost-effectiveness, straightforward synthesis for large-scale production, high adhesion strength, outstanding performance at extremely low temperatures, exceptional resistance to organic solvents, and the capability to prepare multi-functional composite materials.
 - (2) This is the first report that poly(ionic liquid) materials can be simply prepared by neutralization reaction and spontaneous polymerization. Almost 100% yield quantitative synthesis of poly(ionic liquid)s in this work could be advantageous for its various large-scale applications. Such a method of designing poly(ionic liquid) materials has never been achieved before, as most poly(ionic liquid)s are related to tedious synthesis and relatively low reaction yield, which seriously limit their practical application. Considering the widespread attention and promising application prospects of poly(ionic liquid) materials in various fields, including energy storage

and conversion, we posit that the in-situ spontaneous polymerization for poly(ionic liquid) presented in our study can offer crucial inspiration and guidance for other applications.

- (3) The resulting poly(ionic liquid) exhibits durable, long-term, and recyclable adhesion on diverse substrates, including ceramic, epoxy, and stainless steel. Its maximum adhesion strength reaches 16.2 MPa, surpassing that of most catechol and commercially available adhesives. Remarkably, even at the ultra-low temperature of liquid nitrogen ($-196\text{ }^{\circ}\text{C}$), adhesion strengths of 10.0, 5.0, and 6.8 MPa are maintained on ceramic, epoxy (EP), and stainless steel (SS) surfaces, respectively.
 - (4) Because of their unique organic solvents-resistant ability, the poly(ionic liquid)s-adhered joints were insoluble and stable in a variety of common organic solvents, such as n-hexane, dichloromethane, ethyl acetate, tetrahydrofuran, ethanol, acetone, isopropanol, and acetonitrile.
 - (5) The in-situ spontaneous polymerization process we propose to prepare protic poly(ionic liquid) adhesive materials is highly versatile and can be used in many structures. Various hydroxylamine molecules can be used as raw materials to achieve in-situ polymerization to prepare high-performance adhesive materials.
 - (6) The in-situ spontaneous polymerization process can be used to prepare functional composite materials with excellent comprehensive properties in other fields, such as application as a binder component in the battery. During performance testing, it exhibited superior cycle stability and rate performance compared to many reported materials (*Adv. Funct. Mater.* 2023, 33, 2302951; *ACS Energy Lett.* 2023, 8, 1336; *Chem. Eng. J* 2023, 476, 146299).
- As for the toxicity and corrosiveness of alkanol amines, we would like to respond as follows: The harmful properties partially arise from inherent chemical characteristics such as alkalinity and electrophilicity. It is acknowledged that neutralizing alkanolamine solutions with acids to adjust their pH can reduce their corrosiveness (*Ind. Eng. Chem. Res.* 2009, 48, 6486; *Corrosion in Amine Treating Units*. Woodhead Publishing, 2021, <https://doi.org/10.1016/C2020-0-04139-3>). In this study, a simple acid-base mixing reaction in an aqueous solution led to the alkanol amines existing in the form of cationic salt, thus weakening their alkalinity. Neutralizing the alkalinity with acids can result in less toxic salts and reduce the overall toxicity of the solution. Although we did not have the appropriate experimental conditions, it is possible that the toxicity of the product may have been reduced compared to the raw alkanol amines. Meanwhile, this study emphasized that no

hydroxylamine molecule emission was detected during the isothermal in-situ polymerization process, which indicated that alkanol amines were all solidified inside the polymer adhesive by strong intermolecular interactions. This means that even if the adhesive is used for a long time, there will not be any harmful raw alkanolamine leakage.